# RBM15B recognizes H3K79me2 to guide selective m⁶A-modification of mRNA and enhance oncoprotein translation in MLL-r leukemia

Tian-Qi Chen[1,3], Yu-Meng Sun[1,3], Shun-Xin Zhu[1,3], Xiao-Tong Chen[1], Ke-Jia Pu [1], Heng-Jing Huang[1], Qi Pan[1], Jun-Yi Lian[1], Wei Huang[1], Ke Fang[1], Xue-Qun Luo[2], Li-Bin Huang [2✉], Yue-Qin Chen [1✉] & Wen-Tao Wang [1✉]

## Abstract

**The distribution of N⁶-methyladenosine (m⁶A) controls its substrate RNA fate, playing key roles in various biological processes. However, the mechanism underlying site-selective m⁶A deposition of RNAs, especially in the start codon regions, and the role in epigenetic information transduction connecting tumorigenesis remain largely unknown. Here, we identified RBM15B mainly modulates m⁶A modifications in the 5′untranslated regions (UTRs) and around the start codons of mRNAs transcribed. This process is guided by H3K79me2 histone methylation, a critical epigenetic modification in mixed lineage leukemia. We show that the H47 of RBM15B is a key residue for the recognition of H3K79me2. The selective m⁶A modification orchestrated by the H3K79me2–RBM15B axis enhances translation efficiency of oncogenic transcripts, and promotes self-renewal of leukemic stem cells and leukemia maintenance. We further demonstrate that blockade of the H3K79me2-RBM15B-m⁶A axis inhibits the survival of leukemia cells and promotes cell differentiation, and impairs hematological malignancies. This study uncovers a novel selective m⁶A deposition mechanism mediated by H3K79me2 and RBM15B, highlighting promising therapeutic targets for hematological malignancies.**

**Keywords** RBM15B; H3K79me2; N6-methyladenosine; Translation
**Subject Categories** Cancer; Chromatin, Transcription & Genomics; Translation & Protein Quality

## Introduction

N⁶-methyladenosine (m⁶A) modification is the most prevalent epigenetic mark in eukaryotic mRNAs (Jia et al, 2011; Yang et al, 2018; Zaccara et al, 2019; Zeng et al, 2023). It is reversible and affects numerous biological processes, including leukemogenesis (Paris et al, 2019; Roundtree et al, 2017; Vu et al, 2017; Wang et al, 2020a; Weng et al, 2018; You et al, 2023). Inhibiting the m⁶A writer (methyltransferase-like 14, METTL14; methyltransferase-like 3, METTL3) (Vu et al, 2017; Weng et al, 2018), its eraser (fat mass and obesity-associated protein, FTO; AlkB homolog 5, AlkBH5) (Huang et al, 2019b; Li et al, 2017b; Wang et al, 2020a), or its reader (insulin-like growth factor 2 mRNA binding protein 2, IGF2BP2; YTHDF1/2; YTH domain-containing family protein 1 and 2) (Feng et al, 2021; Paris et al, 2019; Weng et al, 2022), can suppress the progression of both hematopoietic stem/progenitor cells and acute myeloid leukemia (AML). Moreover, the deposition of m⁶A on different transcripts or in different regions of a transcript determines targeted RNA fate and affects disease progression (Deng et al, 2018; Einstein et al, 2021; Hu et al, 2022). Previous study illustrated that m⁶A is found on chromatin-associated RNAs, to modulate chromatin state and regulate transcription, and which is methylated in a co-transcriptional process(Barbieri et al, 2017; Liu et al, 2020; Liu et al, 2021; Wei et al, 2022; Xu et al, 2021; Yang et al, 2024). Histone modification H3K36me3 has been reported to mediate METTL14 regulating m⁶A modification enriched near the 3' end region in transcripts(Huang et al, 2019a). Also, RNA-binding protein RBFOX2 could guiding m⁶A modification of promoter-associated RNA (paRNAs) during locus-specific chromatin regulation(Dou et al, 2023). A recent study exhibited that exon junction complexes (EJCs) can protect exon junction–proximal RNA within coding sequences from m⁶A methylation but not in long internal and terminal exons(He et al, 2023). These seminal studies have expanded the understanding of the process in the specific deposition of m⁶A on RNAs in some specific scenarios. However, the mechanisms underlying site-specific selectivity of m⁶A modification and its role in epigenetic information transduction connecting tumorigenesis are still poorly understood.

Many m⁶A sequencing data have shown that, aside from the stop coding region, the start codon region of RNAs also has abundant m⁶A modifications, and mainly regulates translation. However, the specific mechanism of m⁶A-selective deposition in the start codon region of RNA remains exclusive. A recent study has described the

[1]MOE Key Laboratory of Gene Function and Regulation, Guangdong Province Key Laboratory of Pharmaceutical Functional Genes, State Key Laboratory for Biocontrol, School of Life Sciences, Sun Yat-sen University, 510275 Guangzhou, China. [2]Department of Pediatrics, The First Affiliated Hospital, Sun Yat-sen University, 510080 Guangzhou, China. [3]These authors contributed equally: Tian-Qi Chen, Yu-Meng Sun, Shun-Xin Zhu. ✉E-mail: huanglb3@mail.sysu.edu.cn; lsscyq@mail.sysu.edu.cn; wangwt8@mail.sysu.edu.cn

crosstalk between m⁶A and epigenomic layers in human disease (Li et al, 2024). Thus, we asked whether epigenetic modifications, such as histone modifications, function in site-specific selectivity of m⁶A modification, especially in the start codon region of RNAs. To address this question, we choose mixed lineage leukemia, also called *MLL*-rearrangement leukemia (*MLL*-r leukemia), as a model, this type of leukemia has been reported to be driven by significant and specific dysregulation in epigenetic modifications (Okada et al, 2005). The chromosomal translocation of an allele of the *MLL* gene on 11q23 with more than 100 partners to form fusion oncoproteins with a very poor prognosis (Rao and Dou, 2015). MLL-fusion proteins and its chaperonin DOT1L trigger ectopic and higher-order methylation of H3K79 (Bernt et al, 2011; Geng et al, 2012; Yi and Ge, 2022), which is enriched at the transcription start site (TSS) and the first exon and intron (Huff et al, 2010; Perner et al, 2020). These raise a speculation whether H3K79me2/3 are specifically recruited for uniquely selective m⁶A modification in the start codon region of RNA, and such m⁶A site-selectivity is essential for cell fate?

RNA-binding motif protein 15B (RBM15B) was initially identified as a binding partner of the Epstein-Barr virus mRNA export factor and as a cofactor of the nuclear export receptor NXF1 (Hiriart et al, 2005; Uranishi et al, 2009). More recently, it has been shown to play a key role within m⁶A methyltransferase complex (MTC), recruiting the m⁶A MTC to specific RNA regions by recognizing RNA motifs and regulating multiple functions (Huang et al, 2021; Shi et al, 2019). For example, the upregulation of RBM15B induces m⁶A hypermethylation and promotes lung cell differentiation, hepatocellular and prostate carcinoma cell proliferation (Cheng et al, 2025; Dong et al, 2025; Tan et al, 2022). RBM15B can also bind to the regions adjacent to the 5′ end of X-inactive specific transcript (XIST) to mediate m⁶A modification and promote XIST-mediated gene silencing (Patil et al, 2016), suggesting that RBM15B may contribute to the m⁶A distribution of RNAs, though the exact regulatory mechanism behind this remains unknown.

Here, we uncovered that RBM15B, identified as an epigenetic transducer, selectively regulated m⁶A-modified regions in the 5′ UTR and in part of the CDS around the start codon (two wings around start codon (TWINS)) of RNAs, which is guided by H3K79me2. We also found that highly RBM15B is a risk factor for the *MLL*-r leukemia. The selective m⁶A modification guided by H3K79me2–RBM15B axis could regulate translation efficiency of oncogenic transcripts, which is critical for the epigenetic regulation of *MLL* rearrangements in leukemia progression. This study presents a previously unknown mechanism of selective model of m⁶A deposition on TWINS regions of mRNAs, which has potential therapeutic implications in hematopoietic malignancy.

## Results

### RBM15B is a key regulator of uniquely selective m⁶A modification in H3K79 hypermethylation-driven leukemia

To discover the specific factors that might function in the potentially unique selective m⁶A modification, we analyzed the expression level of each component of the m⁶A writer complex in a publicly available dataset with 419 patient samples (Lavallée et al,

2015; Data ref: Lavallée et al, 2015). *RBM15B* was the only one whose expression was greater in the *MLL*-r samples, characterized by H3K79 hypermethylation, compared with the *MLL*-wt samples (Fig. 1A; Appendix Fig. S1A). In a validation set and cell lines, the *RBM15B* mRNA and protein level was greater in *MLL*-r leukemia patients and cell lines than in *MLL*-wt subtypes (Fig. 1B–D; Appendix Fig. S1B and Appendix Table S1). Moreover, we found that highly RBM15B is associated with a poor outcome (Fig. 1E,F; Appendix Table S2). Taken together, the high expression of RBM15B could serve as a critical risk factor and might play a pivotal epigenetic role in *MLL*-rearranged contexts, which harbor H3K79 hypermethylation.

We then examine the regulatory mechanism governing elevated RBM15B expression in *MLL*-r leukemia. Previous studies have revealed that MLL fusion protein mediates transcription regulation mainly depends on the formation of MLL-fusion protein (MLL-FP) complex, including MLL-FP/DOT1L complex-H3K79me axis or MLL-FP/BRD4 complex-H3K27ac axis. We reanalyzed the MLL-FP Cut&Tag, BRD4 and H3K27ac ChIP datasets (Erb et al, 2017; Data ref: Erb et al, 2017; Gilan et al, 2016; Data ref: Gilan et al, 2016; Janssens et al, 2021; Data ref: Janssens et al, 2021), and found that high enrichment of H3K27ac, MLL-FP and BRD4 in *RBM15B* locus, and BRD4 inhibitor (iBET151) can significantly suppress the enrichment of BRD4 on RBM15B locus (Appendix Fig. S1C). These analysis data suggested that MLL-FP/BRD4 complex may regulate RBM15B expression. To validate this hypothesis, we treated the *MLL*-r leukemia cells by iBET151 and then detected the RBM15B expression levels by qRT-PCR and western blot. As shown in Appendix Fig. S1D, iBET151 can significantly inhibit the RBM15B expression. Knockdown of MLL-AF9 can reduce RBM15B expression (Appendix Fig. S1E). ChIP assays also showed that MLL and BRD4 enriched on RBM15B locus and these enrichments can be destroyed by iBET151 (Appendix Fig. S1F). These data may suggest that MLL-FP/BRD4 complex located at the *RBM15B* locus and regulate its expression.

We next investigated the potential effects of RBM15B on m⁶A modification of selected RNA loci. We performed MeRIP-seq and GLORI, which is a transcriptome-wide method that enables absolute quantification of m⁶A at single-base resolution (Liu et al, 2023), in *MLL*-r leukemia cells with or without RBM15B depletion. Both MeRIP-seq and GLORI results showed that the most common m⁶A consensus sequence (GGAC) was enriched, and m⁶A-methylated peaks were mostly present in CDSs (Fig. 1G–J; Appendix Fig. S2A). RBM15B suppression slightly altered m⁶A distribution in total RNA (Fig. 1I,J, left; Appendix Figs. S1G and 2B). When we focused on specific mRNAs from target genes of *MLL*-fusion proteins (Appendix Fig. S2C) (Bernt et al, 2011; Jiang et al, 2012; Wu et al, 2021), we observed a marked decrease in m⁶A in 5′ UTRs and partial CDSs around the gene start codon, as compared with CDSs near the stop codon and 3′ UTRs (Fig. 1I,J, right; Appendix Fig. S2D). We named this region two wings around the start codon (TWINS). At the *MYC*, *HOXA9*, and *BCL2* loci, the representative genes which closely associated with malignant progression of *MLL*-r leukemia, the TWINS region had a high abundance of m⁶A modifications that decreased markedly upon RBM15B knockdown (Fig. 1K). We also demonstrated that RBM15B suppression specifically reduced m⁶A levels in the TWINS region of mRNAs from target genes of MLL-fusion proteins and had no significant effects on the stop codon region (Fig. 1L). These

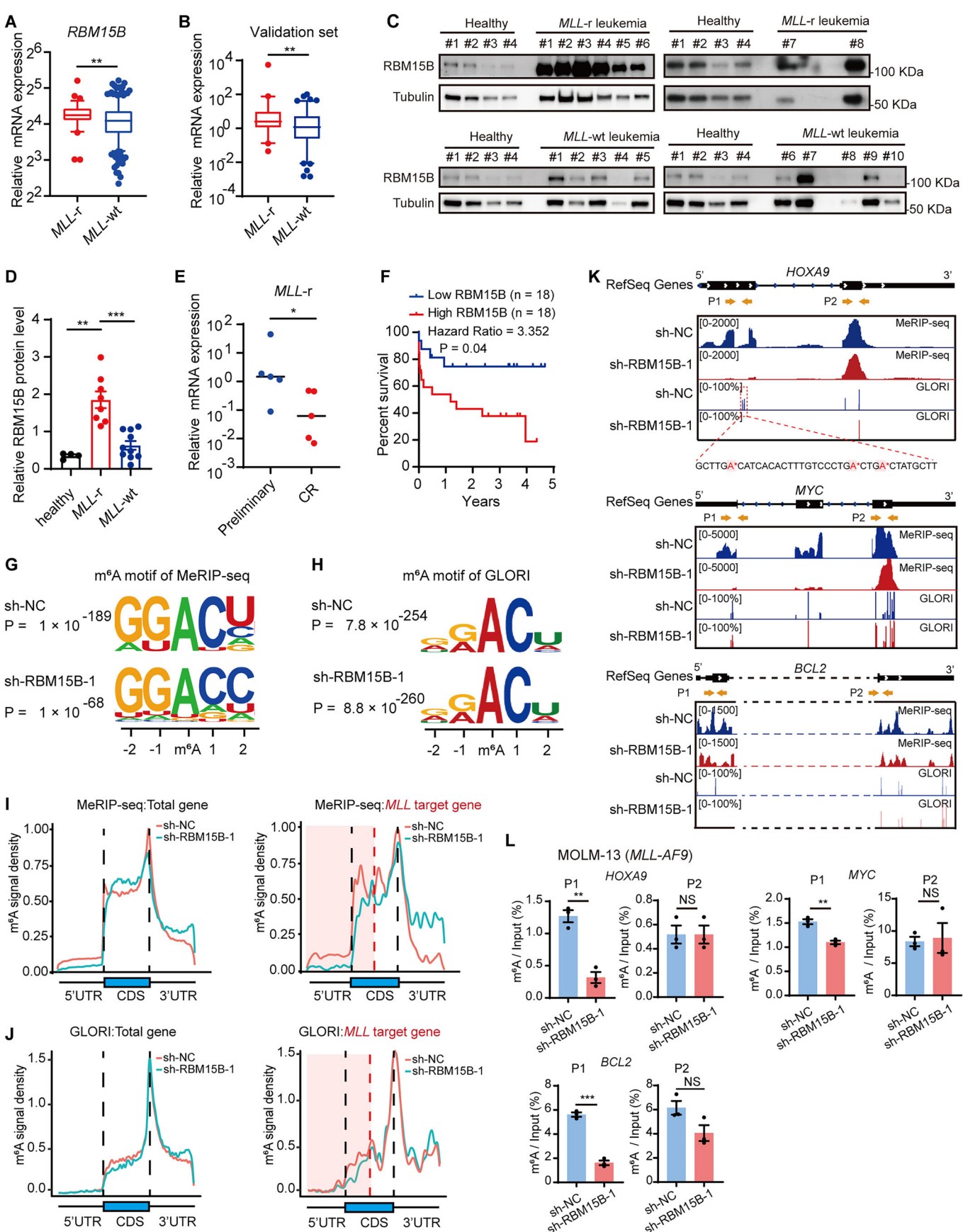

**Figure 1.   RBM15B is highly expressed in *MLL*-r leukemia and mediated TWINS region m⁶A deposition.**

(A) Expression levels of *RBM15B* from the datasets (Lavallée et al, 2015; Data ref: Lavallée et al, 2015). A total of 419 patient samples were classified into *MLL*-r leukemia (n = 35) and *MLL* wild-type (*MLL*-wt) (n = 384) subtypes. Data are presented as median with interquartile range. Each box-and-whisker consists of the 25th quantile (the upper border of the box), median (horizontal line inside the box), 75th quantile (the lower border of the box); whiskers indicate the 5th and 95th percentiles. Mann–Whitney test, P = 0.0098. (B) qRT-PCR analysis was used to determine the expression of *RBM15B* in patient-derived primary leukemia cells of various subtypes, including *MLL*-r leukemia (n = 45) and *MLL*-wt leukemia (n = 108). The Mann–Whitney test was used to determine the significance of the difference in RBM15B mRNA levels between the two groups. Data are presented as median with interquartile range. Each box-and-whisker consists of the 25th quantile (the upper border of the box), median (horizontal line inside the box), 75th quantile (the lower border of the box); whiskers indicate the 5th and 95th percentiles. Mann–Whitney test, P = 0.0019. (C, D) Western blot (C) and relative Protein expression levels (D) of RBM15B in control from healthy donors (n = 4) and primary patients with *MLL*-r leukemia (n = 8) and *MLL*-wt leukemia (n = 10). The Kruskal–Wallis test was implemented when the comparison was among three groups, and multiple comparisons were done using the least significant difference t test after the relative concentration was ranked. Error bars denote means ± SEM. Healthy vs *MLL*-r patient, P = 0.001; *MLL*-wt vs *MLL*-r patient, P = 0.0009. (E) qPCR for RBM15B expression in five pairs of preliminary vs. complete remission (CR) *MLL*-r leukemia patients. GAPDH mRNA as the internal control. Mann–Whitney test, P = 0.0317. (F) The leukemia-free survival of patients with a high expression level of RBM15B is less than that of patients with a low *RBM15B* level in *MLL*-r leukemia. (G, H) Motif analysis revealed N⁶-methyladenosine (m⁶A) motifs enriched in methylated RNA immunoprecipitation sequencing (MeRIP-seq) (G), and glyoxal and nitrite-mediated deamination of unmethylated adenosines-sequencing (GLORI) (H) peaks of control (sh-NC) and RBM15B knockdown (sh-RBM15B) MOLM-13 (*MLL-AF9*) cells. P values were calculated by HOMER. (I, J) Metagene analysis showed the differences in m⁶A signal density distribution between total RNAs (left) and 144 MLL-fusion protein target genes (right) with different treatments (sh-NC or sh-RBM15B), where a marked decrease in m⁶A deposition was observed in the TWINS region after RBM15B knockdown by using MeRIP-seq (I) and GLORI (J). (K) MeRIP-seq and GLORI profiles of m⁶A modification at the *MYC*, *HOXA9*, and *BCL2* genomic loci in RBM15B knockdown compared with control MOLM-13 cells. The y axis scales represent read count of MeRIP-seq, and the m⁶A site along with their methylation level (%) are identified by GLORI. A* indicates the m⁶A site. The positions of Primer 1 (P1) and Primer 2 (P2) related to (L) are shown. (L) Statistical analysis of MeRIP-qPCR data reveals a reduction in m⁶A level at the TWINS region (P1) with *RBM15B* suppression in MOLM-13 (*MLL-AF9*) cells. P = 0.0017 for *HOXA9*; P = 0.0026 for *MYC*; P < 0.0001 for *BCL2*. No significant differences were present in the stop codon region (P2). P > 0.9999 for *HOXA9*; P = 0.8344 for *MYC*; P = 0.0731 for *BCL2*. Error bars denote means ± SEM (n = 3). Two-tailed Student's t test was used to determine statistical significance. NS not significant; * P < 0.5; ** P < 0.01; *** P < 0.001. Source data are available online for this figure.

results suggest that RBM15B may be a key regulator for uniquely selective m⁶A modification, at least in target genes of MLL-fusion proteins.

## RBM15B-regulated m⁶A deposition of the TWINS region mainly depends on H3K79me2

Although a previous study showed that the RNA-binding site bias of RBM15B is U-rich (Patil et al, 2016), we did not find any significant difference in U-rich motif enrichment in the TWINS region compared with other regions (non-TWINS) in transcripts of target genes of MLL-fusion proteins (Appendix Fig. S2E) (Bailey et al, 2015). This result suggests that there is an unrecognized mechanism for RBM15B regulating m⁶A deposition in specific regions. As shown above, the decreased m⁶A sites of sh-RBM15B are mostly distributed in the TWINS region, which coincides with sites of abnormal H3K79 hypermethylation, especially H3K79me2 and H3K79me3 (Deshpande et al, 2014; Vlaming and van Leeuwen, 2016) in *MLL*-rearranged contexts. Therefore, we asked whether RBM15B regulates H3K79me2- or H3K79me3-associated m⁶A deposition in the TWINS region.

Using an analysis between m⁶A distribution obtained from MeRIP-seq/GLORI and H3K79me2/H3K79me3 modification peaks from our previous work (Wang et al, 2020b), we observed that abundant of m⁶A peaks associated with H3K79me2 and H3K79me3 modification in the human genome, respectively (Fig. 2A). At the loci of target genes of MLL-fusion proteins, m⁶A enrichment at sites of H3K79 methylation was significantly increased compared to total gene; especially, m⁶A was more enriched at sites of H3K79me2 than H3K79me3 (Fig. 2A). Distribution analysis also showed an overrepresentation of m⁶A peaks near H3K79me2 modifications (Fig. 2B). These data imply that m⁶A modification of special regions is mainly guided by H3K79me2 modification.

We first used the DOT1L inhibitor EPZ5676 to diminish H3K79me2 genome-wide and found that H3K79me2 cannot regulate the expression of the core components of m⁶A MTC or of RBM15B in *MLL*-r leukemia cells (Appendix Fig. S2F,G). We also determined that H3K79me2 suppression slightly affected m⁶A distribution in total RNAs (Appendix Fig. S3A). By contrast, H3K79me2 suppression caused a significant decrease in m⁶A in the TWINS region of transcripts of target genes of MLL-fusion proteins, similar to the effects of RBM15B knockdown (Fig. 2C; Appendix Fig. S3B,C). In addition, MeRIP-seq and chromatin immunoprecipitation-sequencing (ChIP-seq) data (Perner et al, 2020; Data ref: Perner et al, 2020) showed that 65 target genes of MLL-fusion proteins displayed a decrease in m⁶A and H3K79me2 modifications after both RBM15B knockdown and EPZ5676 treatment compared to untreated controls (Fig. 2D; Appendix Fig. S3D), indicating that H3K79me2 participates in the regulation of RBM15B-mediated m⁶A modification in transcripts of target genes of MLL-fusion proteins.

As expected, suppression of H3K79me2, both by EPZ5676 treatment and DOT1L knockdown, reduced m⁶A levels of transcripts of target genes of MLL-fusion proteins (Fig. 2E; Appendix Fig. S3E–K). m⁶A levels decreased only in *MLL*-r leukemia cell lines but not in *MLL*-wt cell lines, in which H3K79me2 is not indispensable and presents low abundance (Fig. 2F; Appendix Fig. S3H). RBM15B knockdown did not show any influence on H3K79me2 modification in the corresponding gene loci, implying that RBM15B might be downstream of H3K79me2 (Appendix Fig. S3L,M). However, overexpression of RBM15B in H3K79me2-depleted cells could not rescue the m⁶A decrease, implying that RBM15B regulation of m⁶A modification could be at least partly dependent on H3K79me2 (Fig. 2G). Together, our data suggest that RBM15B is a key regulator of m⁶A modification via H3K79me2–m⁶A crosstalk in the *MLL*-rearrangement context.

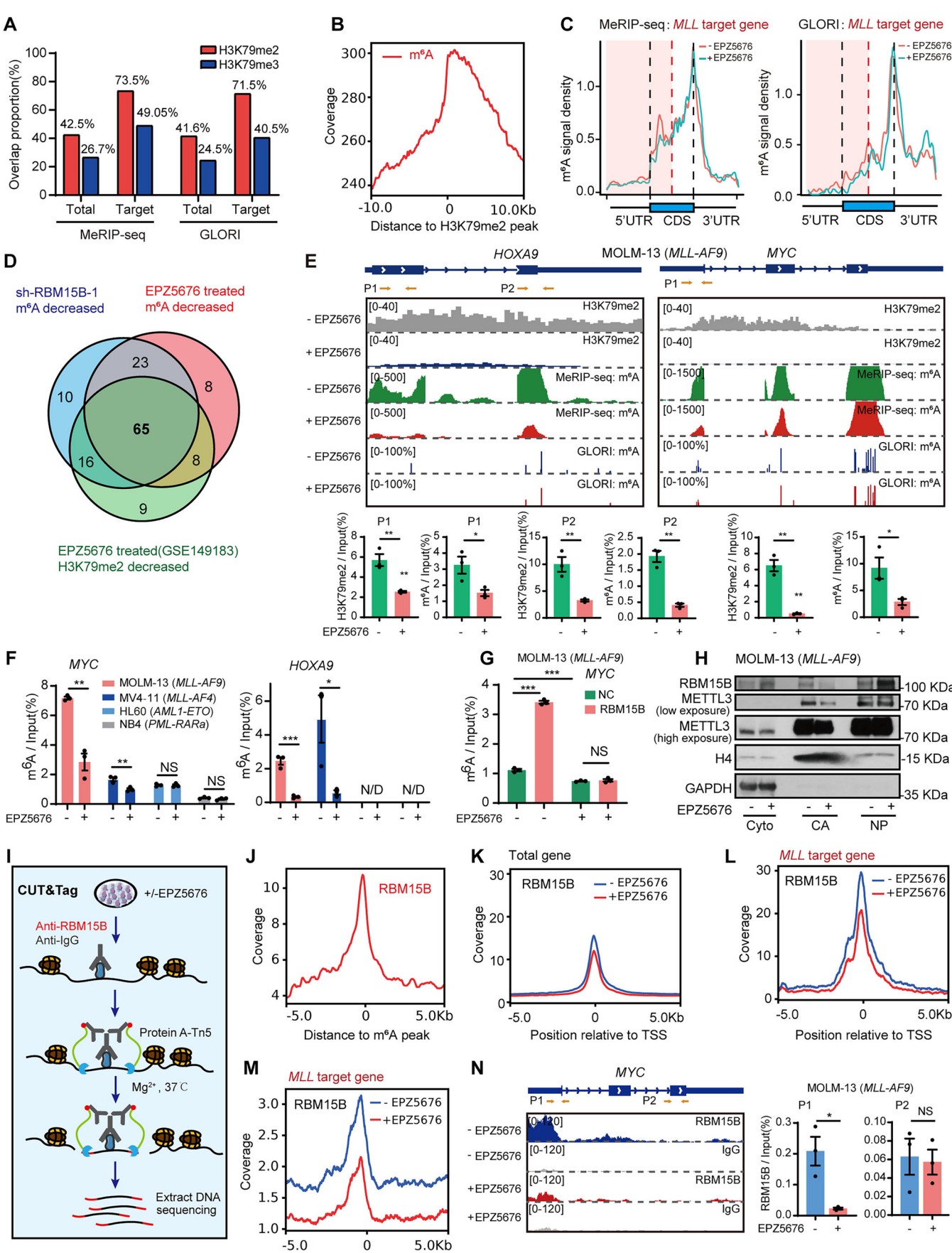

**Figure 2.  H3K79me2 guides RBM15B-regulated m⁶A deposition of the TWINS region.**

(A) Overlap of m⁶A peaks with H3K79me2 and H3K79me3 marks in the human genome. The histogram shows that the H3K79me2 modification sites had greater overlap with m⁶A peaks than did H3K79me3 sites. Our previous work ChIP-seq data (Wang et al, 2020b; Data ref: Wang et al, 2020b), and our MeRIP-seq and GLORI data obtained from MOLM-13 cells were used for the analyses. (B) Distribution of m⁶A modifications to H3K79me2 peaks in MOLM-13 (*MLL-AF9*) cells. (C) Metagene analysis of m⁶A signal density in *MLL*-r target genes by using MeRIP-seq (left) and GLORI (right). MOLM-13 (*MLL-AF9*) cells were treated with or without EPZ5676. (D) Venn diagrams showing that 65 MLL-fusion protein target genes decreased m⁶A and H3K79me2 modifications in both RBM15B knockdown and EPZ5676-treated cells. (E) Genome browser views of H3K79me2 (ChIP-seq) (Perner et al, 2020; Data ref: Perner et al, 2020) and m⁶A (MeRIP-seq and GLORI-seq) signals for *HOXA9* and *MYC* in MOLM-13 (*MLL-AF9*) cells treated with or without EPZ5676 (top) and verified by ChIP- and MeRIP-qPCR, respectively (bottom). The positions of Primer 1 (P1) and Primer 2 (P2) are shown. (F) MeRIP-qPCR for *MYC* and *HOXA9* in MOLM-13 (*MLL-AF9*), MV4-11 (*MLL-AF4*), HL60 (*AML1-ETO*), and NB4 (*PML-RARα*) cell lines treated with or without EPZ5676. N/D, no signal in the qPCR detection. $N = 3$ biological replicates. The *P* values for *MYC* (left to right): 0.0017, 0.0099, 0.8797, and 0.3151; The *P* values for *MYC* (left to right): 0.0007, 0.0363, N/D, and N/D. (G) MeRIP-qPCR for *MYC* in control (NC) and RBM15B-FLAG overexpressing MOLM-13 (*MLL-AF9*) cells treated with or without EPZ5676. NC (without EPZ5676) vs RBM15B-FLAG (without EPZ5676), $P < 0.0001$; NC (without EPZ5676) vs NC (with EPZ5676), $P = 0.0009$; NC (with EPZ5676) vs RBM15B-FLAG (with EPZ5676), $P = 0.5739$. (H) Western blot for RBM15B and METTL3 in subcellular fractions of MOLM-13 (*MLL-AF9*) cell, including Cyto, CA, and NP. GAPDH and H4 acted as the cytoplasmic and nucleoplasmic controls, respectively. Cyto cytoplasm, CA chromatin-associated fraction; NP: nucleoplasmic fraction. (I) Schematic illustration of the RBM15B CUT&Tag followed by high-throughput sequencing analyses. The global DNA-binding sites of RBM15B were detected by anti-RBM15B. Using IgG as a negative control. (J) Distribution of m⁶A modifications to the RBM15B DNA-binding peaks. (K, L) Meta-analysis plot showing the RBM15B binding profiles across the +5 kb to −5 kb genomic region around the transcription starts site (TSS) of total genes (K) and *MLL*-r target genes (L). Profiles of EPZ5676 treatment (red) compared with control (blue) MOLM-13 cells are presented. (M) Distribution of RBM15B binding sites with or without EPZ5676 treatment to the nearest H3K79me2 (Wang et al, 2020b; Data ref: Wang et al, 2020b) peaks. (N) Genome browser views of RBM15B (CUT&Tag-seq) and IgG signals for *MYC* in MOLM-13 treated with or without EPZ5676 (left) and verified by ChIP-qPCR (right). The positions of Primer 1(P1) and Primer 2(P2) are shown. $P = 0.0162$ for P1, $P = 0.8164$ for P2. Values are mean ± SEM ($n = 3$), one-way ANOVA in (G), and unpaired two-tailed Student's *t* test in (E, F, N). NS not significant; *$P < 0.05$; **$P < 0.01$; ***$P < 0.001$. Source data are available online for this figure.

## H3K79me2 guides RBM15B to the TWINS region in chromatin

We next investigated the role of RBM15B regulation of the uniquely selective m⁶A modification in the *MLL*-rearranged context through H3K79me2. We firstly confirmed RBM15B partly existed in the chromatin subcellular fraction (Appendix Fig. S4A). When we inhibited the H3K79me2 modification, less RBM15B and METTL3 bound to chromatin (Fig. 2H; Appendix Fig. S4B), suggesting that retention of RBM15B and the m⁶A MTC on chromatin might partially depend on H3K79me2 modification.

We next performed a CUT&Tag analysis, which can be used to precisely map the global DNA-binding sites of RBM15B (Fig. 2I). Distribution analysis showed an overrepresentation of m⁶A peaks near RBM15B-binding sites, indicating that the location of RBM15B in DNA is important for selective m⁶A modification (Fig. 2J). RBM15B-binding sites were enriched near the transcription start site (TSS) (Fig. 2K,L). Previous studies have illustrated that H3K79me2 is distributed at 5′ UTR, TSS and the first exon and intron, and the peak of H3K79me2 is just behind the TSS and gradually declines throughout the first intron (Appendix Fig. S4C,D) (Huff et al, 2010; Perner et al, 2020). The distribution analysis showed an overrepresentation of RBM15B-binding sites near H3K79me2 (Wang et al, 2020b; Data ref: Wang et al, 2020b) modifications peaks (Fig. 2M). Moreover, the global levels of RBM15B and H3K79me2 (Wang et al, 2020b; Data ref: Wang et al, 2020b) enrichment signal showed a positive correlation across the human genome, especially the MLL-fusion protein target genes (Appendix Fig. S4F,G). After H3K79me2 depletion, both H3K79me2 and RBM15B binding significantly decreased, especially at the TSS loci of target genes of MLL-fusion proteins (Fig. 2K,L; Appendix Fig. S4C,D). These data may suggest a role for H3K79me2 in RBM15B recruitment at the TWINS region of target genes.

We also performed ChIP-qPCR assays and established that RBM15B bound to *HOXA9*, *BCL2*, and *MYC* gene loci (Appendix Fig. S4H,I). When we decreased H3K79me2, we observed decreased RBM15B binding to the TWINS region of *MYC* and *BCL2* chromatin, where H3K79me2 enrichment was decreased; while there were no significant changes in RBM15B binding in the stop codon regions of these loci, where with low H3K79me2 enrichment (Fig. 2E,N; Appendix Figs. S3G and 4J). In addition, EPZ5676 treatment triggered a significant decrease in RBM15B binding to chromatin at both the TWINS and stop codon regions of the *HOXA9* gene locus, which harbored high levels of H3K79me2 modification (Fig. 2E; Appendix Fig. S4J). These findings suggest that H3K79me2 is required for RBM15B-guided m⁶A MTC binding to specific target chromatin regions.

## RBM15B guides selective m⁶A modification by directly binding H3K79me2 and recruiting the MTC

We further endeavored to uncover the regulatory mechanism of RBM15B in modulating H3K79me2–m⁶A crosstalk. Colocalization and Co-IP analysis showed the correlation between RBM15B and H3K79me2 modification in the nucleus (Fig. 3A,B; Appendix Fig. S5A), and this correlation is nucleic acid–independent (Appendix Fig. S5B). In addition, RBM15B promoted METTL3 binding to H3K79me2 modification (Fig. 3C,D). RBM15B knockdown significantly reduced METTL3 enrichment on *HOXA9*, *BCL2*, and *MYC* gene loci (Fig. 3E). Together, these results suggest that RBM15B-mediated selective m⁶A modification requires higher-order H3K79me2 followed by m⁶A MTC recruitment.

A recent study identified that MEN1 is a reader of H3K79me2 (Lin et al, 2023); however, a MeRIP assay suggested that MEN1 did not participate in m⁶A deposition on target genes of MLL-fusion proteins (Appendix Fig. S5C,D). Therefore, we explored whether RBM15B could directly function as an anchor linking the m⁶A MTC to H3K79me2. An in vitro pull-down assay showed that RBM15B could directly interact with the H3K79me2 but not interact with H3.3 (Fig. 3F; Appendix Fig. S5E,F). Moreover, we found that RBM15B and METTL3 interacted with the MLL-fusion protein (Appendix Fig. S5G,H). We further reanalyzed the genome-wide occupancy of MLL-AF9 or MLL-AF4 (Janssens et al, 2021;

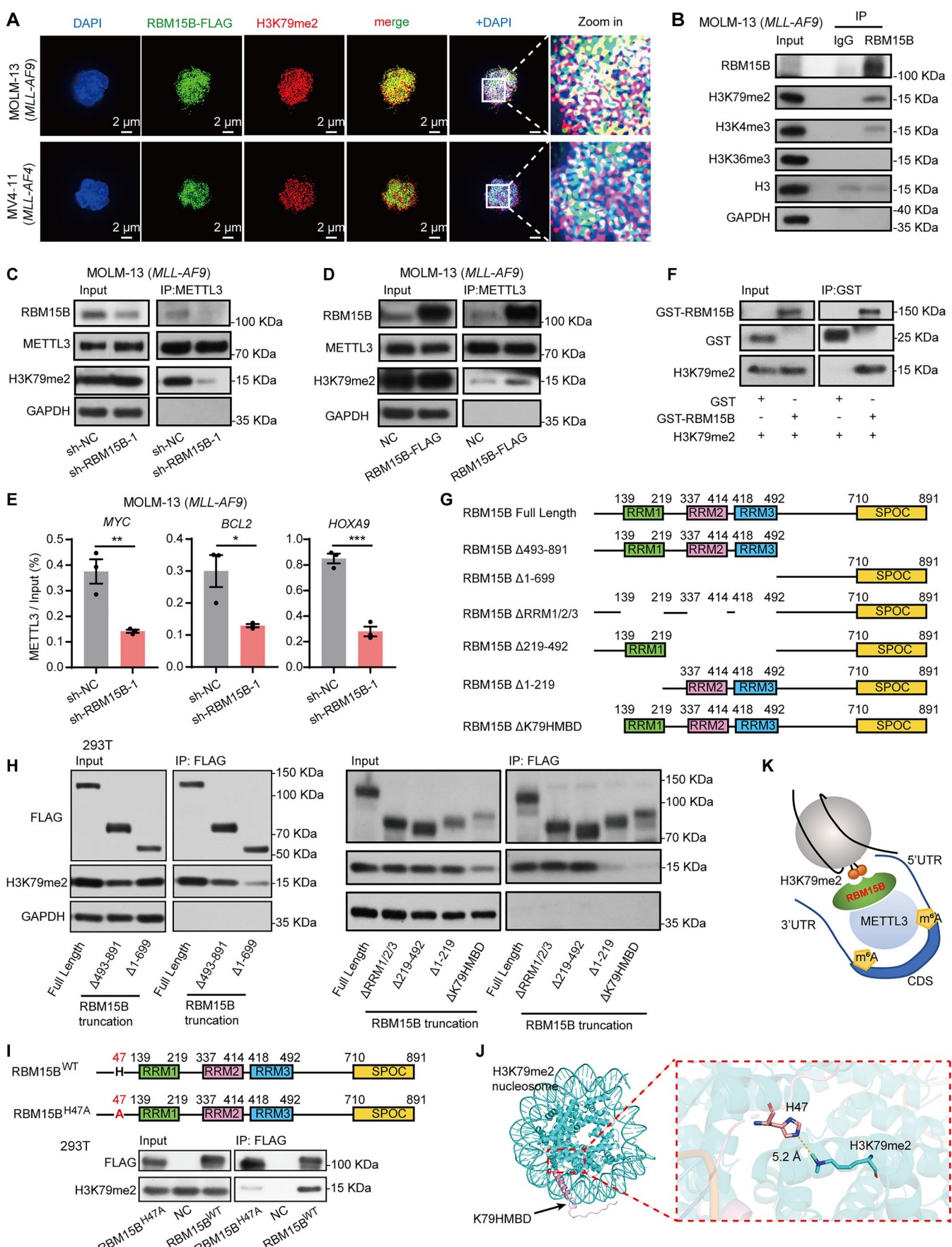

**Figure 3.  K79HMBD is a novel domain required for direct binding to H3K79me2.**

(A) Immunofluorescence of RBM15B-FLAG (green) colocalized with H3K79me2 (red) in MOLM-13 (*MLL-AF9*) and MV4-11 (*MLL-AF4*) cells by using Multi-modality Structured illumination Super resolution Microscopy. Nuclei were stained with DAPI (blue). White plots in the zoom-in picture showed the colocalization site of RBM15B and H3K79me2 in nuclei. Scale bars, 2 µm. (B) Co-immunoprecipitation (Co-IP) assay of RBM15B with H3K79me2 and H3K4me3 in MOLM-13 (*MLL-AF9*) cells. H3K36me3 and H3 showed no interaction with RBM15B. GAPDH as the negative control. (C, D) Co-IP of METTL3 with H3K79me2 in *RBM15B* knockdown (sh-RBM15B-1) (C) or overexpression (RBM15B-FLAG) (D) in MOLM-13 (*MLL-AF9*) cells. (E) METTL3 chromatin immunoprecipitation (ChIP)-qPCR for *MYC*, *BCL2*, and *HOXA9* in the RBM15B knockdown (sh-RBM15B, red) compared with control (sh-NC, gray) in MOLM-13 (*MLL-AF9*) cells. Values are mean ± SEM (*n* = 3) and unpaired two-tailed Student's *t* test; *P* = 0.0082 for *MYC*; *P* = 0.0272 for *BCL2*; *P* = 0.0004 for *HOXA9*. \*\**P* < 0.01; \*\*\**P* < 0.001. (F) Western blot showing an in vitro GST pull-down assay with purified GST-RBM15B and recombinant histone H3K79me2. (G) Schematic for RBM15B-domain deletion mutants. (H) Co-IP of H3K79me2 with FLAG-tagged RBM15B-domain deletion mutants in 293 T cells. Δ, deletion of the indicated domain; RRM1/2/3, FLAG-tagged RRM domain combination as indicated. (I) Co-IP of H3K79me2 with FLAG-tagged RBM15B with H47A mutation or wild-type in 293T cells. H47A, the H47 residue of RBM15B was mutated to alanine; WT, wild-type full-length RBM15B. (J) Molecular docking between the reported H3K79me2 nucleosome and K79HMBD structure using Alphafold3 (Docking score: −244.09 kcal/mol). The H47 of the K79HMBD motif is poised close (5.2 Å) to the H3K79me2. (K) A proposal that RBM15B recruits the m⁶A MTC to specific gene loci for selective m⁶A modification of RNA by directly interacting with H3K79me2. Source data are available online for this figure.

Data ref: Janssens et al, 2021) and m⁶A methylomes. Based on the meta-plot analysis, we observed that both MLL-AF9 and MLL-AF4 enrichment signals near the m⁶A peaks of the transcriptome (Appendix Fig. S5I). These results suggested that MLL fusion protein/H3K79me2 guides RBM15B–MTC to *MLL*-r targeted transcripts for m⁶A modification.

We further identified that known RNA-binding domains of RBM15B, such as RRM1, RRM2, RRM3, and SPOC, did not interact with H3K79me2, but a segment located before RRM1 did (Fig. 3G,H). The novel functional segment was named K79HMBD (for H3K79 hypermethylation binding domain). K79HMBD was located very close to RRM1, RRM2, and RRM3, which could assist RBM15B in simultaneously binding both H3K79me2 and mRNA for m⁶A modification on nearby sites. We further identified key residues of K79HMBD potential for the recognition of H3K79me2. A latest study reported that Menin was a reader of H3K79me2, and the H433 residue of Menin is a key residue for the recognition of H3K79me2 (Lin et al, 2023). We found that K79HMBD region is similar to the Menin fingers domain, and the H47P48 residues of K79HMBD are similar to the H433V434 (hydrophobic amino acid) residues of Menin. We speculated that H47 may be the key amino acid to recognize H3K79me2. Interestingly, Co-IP results showed that RBM15B^H47A slightly binds to H3K79me2, suggesting that RBM15B might be dependent on H47 residue in K79HMBD domain to bind to H3K79me2 and mRNA for m⁶A modification on nearby sites (Fig. 3I). Further, we also did the molecular docking between the reported H3K79me2 nucleosome and K79HMBD structure. Although the K79HMBD structure is hard to predict well by Alphafold3, the successful docking between these molecules is clearly shown (Fig. 3J; Docking score: −244.09 kcal/mol). The H47 of the K79HMBD motif is poised close (5.2 Å) to the H3K79me2 (Fig. 3J). Overall, these findings illustrate that RBM15B recognizes H3K79me2 to recruit the m⁶A MTC to TWINS region of *MLL*-r targets, which guides the m⁶A-selective modification (Fig. 3K).

## Interrupting the H3K79me2-RBM15B-m⁶A axis inhibits leukemia cell self-renewal and promotes cell differentiation

Considering the specific effect of H3K79me2-RBM15B axis on m⁶A-selective modification, we next investigated the function of this pathway in leukemia cells. We block this m⁶A-selective modification axis by targeting RBM15B, which has been proven

to be the transducer of this pathway. Silencing of RBM15B exhibited significant anti-leukemic effects on *MLL*-r leukemia cell lines but slight on *MLL*-wt cell lines, including inhibiting proliferation, cell cycle progression, and promoting apoptosis (Fig. 4A; Appendix Fig. S1H,I and S6A–D), implying a significant anti-cancer role by blocking this H3K79me2-RBM15B-m⁶A axis. These results supported the proposal that H3K79me2-RBM15B-m⁶A axis plays an important role in *MLL*-rearranged contexts, which harbor H3K79me2 hypermethylation, and also explained that this pathway is not indispensable in *MLL*-wt cell lines, which are characteristic of H3K79me2 hypomethylation.

As leukemia is an intractable haematopoietic malignancy, featured by the clonal expansion and differentiation blockage of hematopoietic stem and progenitor cells (DiNardo et al, 2023). We asked whether H3K79me2-RBM15B-m⁶A axis is involved in regulating leukemia cell self-renewal and differentiation events. Interestingly, suppression of H3K79me2-RBM15B-m⁶A axis by RBM15B knockdown significantly reduced colony-forming ability and promoted differentiation of *MLL*-r leukemia cell lines, but had a limited influence on *MLL*-wt cells (Fig. 4B–E; Appendix Fig. S6E–H). While RBM15B overexpression could rescue the cell differentiation triggered by RBM15B knockdown (Fig. 4F; Appendix Fig. S6I), which furthermore confirmed the role of RBM15B in the differentiation blockage of leukemia. The consistent oncogenic role of RBM15B was further demonstrated in primary cells from leukemia patient samples (Fig. 4G; Appendix Fig. S7A–C). These data could suggest RBM15B is required for leukemia cell self-renewal and differentiation events. To further demonstrate the role of H3K79me2-RBM15B-m⁶A axis in the self-renewal of leukemic stem cells (LSCs), we have also purified CD34+ cells from patients suffering *MLL-AF4* leukemia to further investigate the role of RBM15B on leukemia stem cells (LSC) (Fig. 4H). The results showed that RBM15B knockdown attenuated colony-forming unit and replating ability in leukemic CD34+ cells (Fig. 4I). RBM15B knockdown resulted in higher amounts of apoptotic cells in the LSC (Fig. 4J; Appendix Fig. S7D). These results indicated that depletion of H3K79me2-RBM15B-m⁶A axis inhibits self-renewal and promotes cell differentiation at least in leukemia with *MLL*-rearranged context.

## H3K79me2-RBM15B guides selective m⁶A modification of TWINS promotes oncoprotein translation to drive tumorigenesis

To better characterize how the H3K79me2-RBM15B-m⁶A axis regulates the anti-leukemic effects, we subsequently integrated and

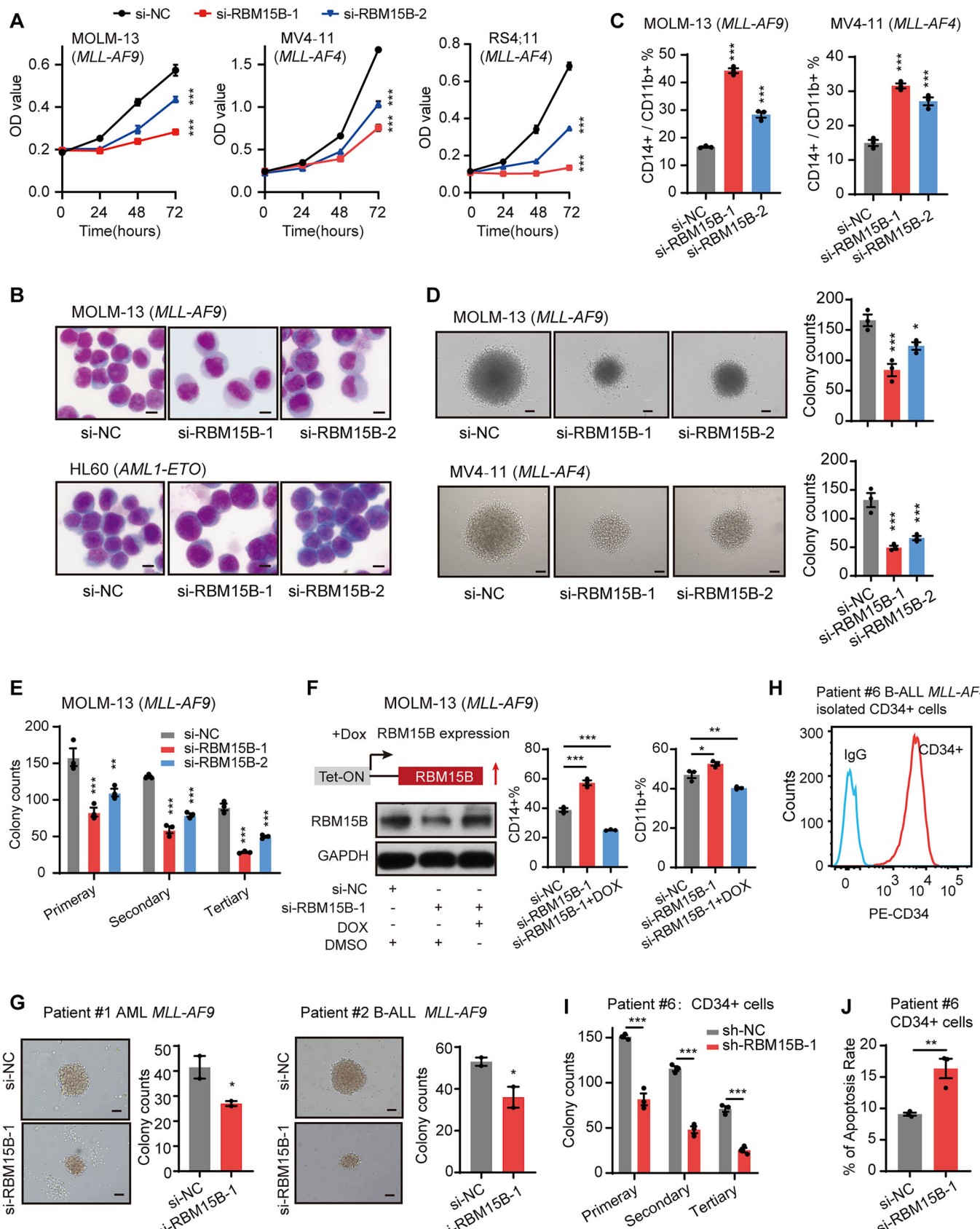

◀ **Figure 4.  RBM15B depletion inhibits leukemia cell self-renewal and promotes cell differentiation.**

(A) Proliferation of *MLL*-r leukemia cell lines (MOLM-13, MV4-11, RS4;11) electrotransfected with si-RNA control (si-NC) or RBM15B small interfering RNA (si-RBM15B). All of the *P* values: *P* < 0.0001 (si-NC vs si-RBM15B-1) *P* < 0.0001 (si-NC vs si-RBM15B-2). (B) Representative images of Wright-Giemsa-stained cells to differentiate morphology in MOLM-13 (*MLL-AF9*) and HL60 (*AML1-ETO*) cell lines. Scale bars, 20 mm. (C) Statistical analysis of flow cytometry data of myeloid markers CD11b and CD14 in MOLM-13 (*MLL-AF9*) and MV4-11 (*MLL-AF4*) cell lines electrotransfected with si-NC or si-RBM15B. All of the *P* values: *P* < 0.0001 (si-NC vs si-RBM15B-1), *P* < 0.0001 (si-NC vs si-RBM15B-2). (D) Representative images and statistical analysis of colony-forming cell (CFC)that tested the in vitro self-renewal potential in MOLM-13 (*MLL-AF4*) and MV4-11(*MLL-AF4*) cell lines electrotransfected with si-NC or si-RBM15B. Scale bars, 50 μm. *P* values for MOLM-13: *P* = 0.0006 (si-NC vs si-RBM15B-1), *P* = 0.0146 (si-NC vs si-RBM15B-2). *P* values for MV4-11: *P* = 0.0003 (si-NC vs si-RBM15B-1), *P* = 0.0008 (si-NC vs si-RBM15B-2). (E) MOLM-13 (*MLL-AF9*) cell electrotransfected with si-NC or si-RBM15B serially replated on cytokine-supplemented methylcellulose (MethoCult™ H4434 Classic) that supports the growth of cell colony-forming units (CFUs). The *P* values are (left to right): 0.0007, 0.0063, <0.0001, <0.0001, <0.0001, and <0.0001. (F) Rescue of RBM15B expression in RBM15B-knockdown MOLM-13 (*MLL-AF9*) cells by doxycycline (Dox)-inducible RBM15B lentivirus expression vector with tetracycline (Tet)-ON promoter. Immunoblot of RBM15B protein level and statistical analysis of flow cytometry analysis of CD11b and CD14 in MOLM-13 (*MLL-AF9*) cells. *P* values for CD14: *P* < 0.0001 (si-NC vs si-RBM15B-1), *P* = 0.0002 (si-NC vs si-RBM15B-1 + DOX). *P* values for CD11b: *P* = 0.0134 (si-NC vs si-RBM15B-1), *P* = 0.0059 (si-NC vs si-RBM15B-1 + DOX). (G) Representative images and statistical analysis of colony formation for *MLL*-r leukemia patient primary cells electrotransfected with si-NC or si-RBM15B-1. Scale bars, 50 μm. AML acute myelocytic leukemia, ALL acute lymphocytic leukemia. *MLL-AF9* indicates *MLL*-fusion genes. *P* = 0.0434 for Patient #1; *P* = 0.0455 for Patient #2. (H) Representative images of flow cytometry for CD34+ cells derived from a *MLL*-r leukemia patient sample (B-ALL, *MLL-AF4*). (I) Serial replating colony formation assay showing the clonogenic defect of primary CD34+ cells isolated from a patient sample (B-ALL, *MLL-AF4*) after RBM15B Knockdown. The *P* values are (left to right): 0.0005, 0.0001, and 0.0003. (J) Percentages of apoptotic CD34+ cells purified from patient sample (B-ALL, *MLL-AF4*) after electrotransfection. *P* = 0.0098. Values are mean ± SEM (*n* = 3), two-way ANOVA in (A), one-way ANOVA in (C–F), and unpaired two-tailed Student's *t* test in (G, I, J). * *P* < 0.05; **P* < 0.01; ***P* < 0.001. Source data are available online for this figure.

analyzed CUT&Tag, ChIP-seq (Perner et al, 2020; Data ref: Perner et al, 2020), and m6A-seq data. The results revealed that 1537 genes displayed a decrease in the m6A marks, H3K79me2 modification, and RBM15B binding after both RBM15B knockdown and EPZ5676 treatment compared to untreated controls (Appendix Fig. S8A). 64 genes were reported targets of MLL-fusion proteins (Fig. 5A), including *MYC*, *HOXA9*, *BCL2*, etc., which are the representative oncogenes contributing to *MLL*-r leukemia progression. RIP-qPCR analysis further verified that H3K79me2 depletion reduced RBM15B and METTL3 interaction with the mRNAs of target genes in the *MLL*-r leukemia cells but not in the *MLL*-wt cells (Fig. 5B,C; Appendix Fig. S8B). All the genes display downregulation enrichments of m6A modification, RBM15B binding, and H3K79me2 modification. These findings suggest that H3K79me2 specifically recruits the RBM15B-guided MTC to target mRNA in an *MLL*-rearranged context. Gene Ontology (GO) analysis revealed that the modification downregulated genes were enriched in regulation of cell differentiation, cell cycle, RNA methylation, and mRNA processing (Appendix Fig. S8C), which is consistent with the function of RBM15B in leukemia progression.

As m6A modification in the 5′ UTR can facilitate translation by recruiting ribosomes (Barbieri et al, 2017; Choe et al, 2018; Meyer et al, 2015), we speculated that the H3K79me2–RBM15B–m6A axis could affect the translation efficiency of target genes. To address this issue, we performed ribo-seq and RNA-seq in the RBM15B-downregulated MOLM-13 cells. We found that downregulation of RBM15B did not affect the mRNA levels of most of m6A-modified transcripts, but reduced translational efficiency of reported *MLL*-r targets (e.g., *HOXA9*, *BCL2*, *MYC*) by 50%, versus 18.4% in non-*MLL*-r targets (Fig. 5D; Appendix Fig. S8D–F). This analysis suggested that RBM15B has a more specific regulatory function on translational efficiency of *MLL*-r targets, but not on RNA transcription level regulation. As expected, the translation efficiency shown by polysome/sub-polysome ratio also confirmed these targets upon RBM15B knockdown and EPZ5676 treatment (Fig. 5E–G; Appendix Fig. S9A–C). In addition, RBM15B knockdown or overexpression could influence the protein levels, but only slight on the distribution, stability, or expression, of the targeted transcripts of MLL-fusion protein (Fig. 5H,I; Appendix

Fig. S9D–G). These targets were further validated in RBM15B-depleted primary *MLL*-r and *MLL*-wt leukemia cells (Fig. 5J).

To further verify the direct role of TWINS m6A modification on translation, we constructed two *GFP* reporter systems, one is *TWINS*m6Awt-*GFP* (DNA sequences containing the native promoter and TWINS region with m6A sites from *HOXA9* or *MYC* fused to the GFP sequence), and another one is *TWINS*m6Amut-*GFP* (DNA sequences containing the native promoter and TWINS region with m6A mutation sites from *HOXA9* or *MYC* fused to the GFP sequence) (Fig. 5K). The m6A sites were identified by GLORI. These *GFP* reporter systems were integrated into DNA via lentivirus in MV4-11 or MOLM-13 cells. We found that *TWINS*m6Awt-*GFP* of *HOXA9* or *MYC* constructs displayed significantly GFP+ signal than control in the flow cytometry results (the mean fluorescence intensity [MFI] as shown) (Appendix Fig. S10A), showing the success of *GFP* reporter system for use to indicate GFP translation in *TWINS*m6Awt-*GFP* of *HOXA9* or *MYC* constructs. We also found slight changes in *GFP* RNA levels between the cells with WT and mutant reporters (Appendix Fig. S10B). To conduct a more accurate quantitative analysis of the differences of GFP translation efficiency, we normalized the GFP-MFI to the relative RNA levels (GFP-MFI/relative expression of mRNAs, GFP-MFI/RE-RNAs) and reassessed the translational output.

As shown in Fig. 5L and Appendix Fig. S10C,D, *TWINS*m6Awt-*GFP* cells exhibited higher GFP translation levels than that of *TWINS*m6Amut-*GFP* cells, supporting the role of m6A modifications within the RNA's TWINS region in enhancing translation (We also found that upregulating RBM15B significantly enhanced GFP translation in *TWINS*m6Awt-*GFP* cells, while downregulating RBM15B reduced it (Fig. 5L,M; Appendix Fig. S10C–J). However, modulating RBM15B expression had a limited influence on GFP translation in *TWINS*m6Amut-*GFP* cells. These findings suggest that m6A within the TWINS region plays an important role in the translation of oncogenic transcripts and that RBM15B selectively facilitates this process. Previous studies have demonstrated that EPZ5676 decreases H3K79me2 modification at the loci of *MLL*-r target genes, and reduces their transcription levels and expression (Stein et al, 2018; Yi and Ge, 2022). We also found that EPZ5676 significantly reduced the MFI of GFP signals in both

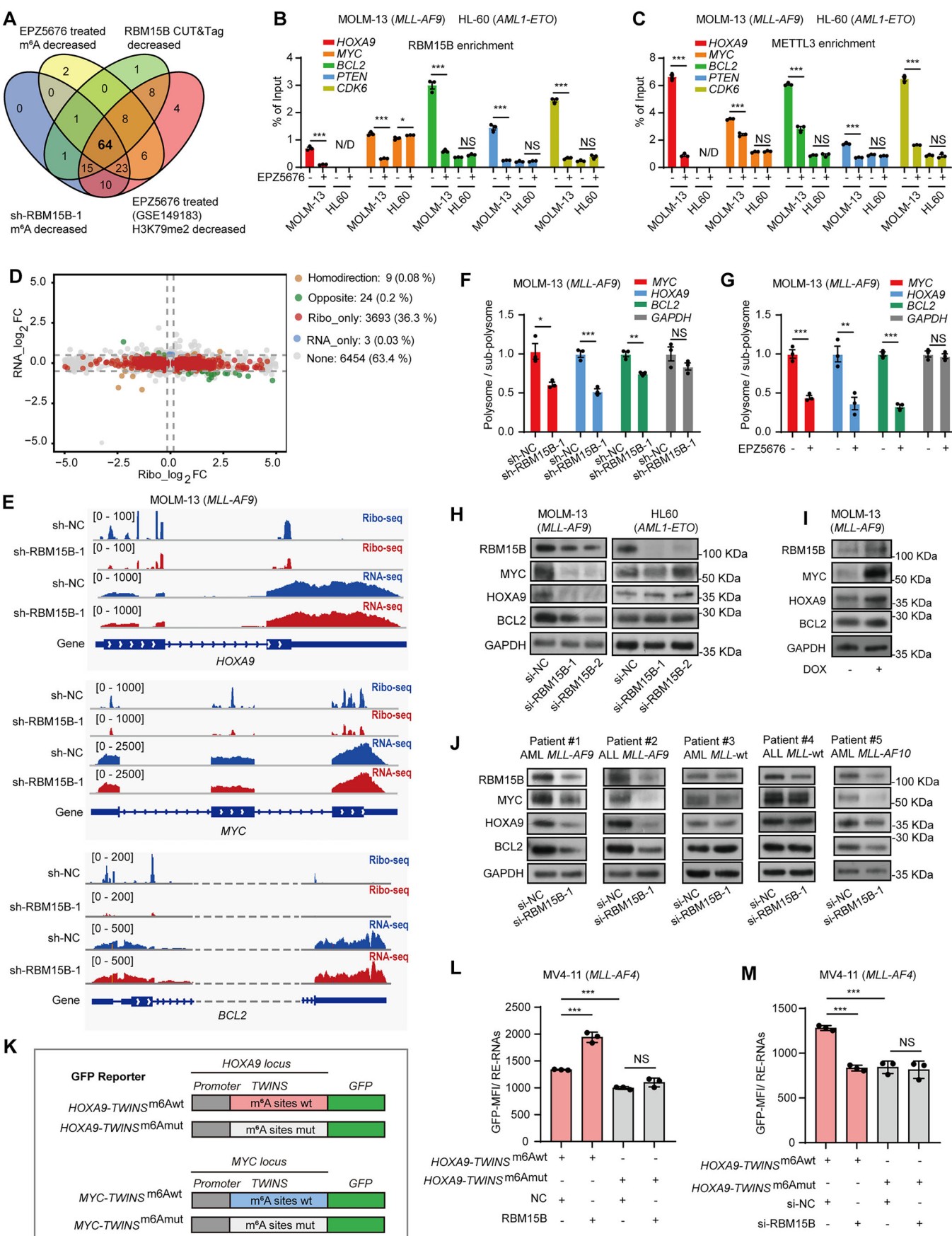

**Figure 5. RBM15B facilitates oncoprotein translation to promote leukemogenesis.**

(A) Venn diagram showing the overlap in sets of target genes of MLL-fusion proteins displaying reduction of H3K79me2, RBM15B, and m6A marks upon integrated analysis of CUT&Tag, ChIP-seq (Perner et al, 2020; Data ref: Perner et al, 2020) and m6A-seq data revealed in both *RBM15B* knockdown (sh-RBM15B) and EPZ5676 treatment compared to the negative control sample. (B, C) RBM15B (B) and METTL3 (C) RIP-qPCR for *MYC, HOXA9, BCL2, PTEN,* and *CDK6* in MOLM-13 (*MLL-AF9*) or HL60 (*AML1-ETO*) cells treated with or without EPZ5676. N/D, no signal in the qPCR detection. The *P* values for (B) (left to right): 0.0002, N/D, < 0.0001, 0.0164, <0.0001, 0.0544, <0.0001, 0.4894, <0.0001 and 0.0536; The *P* values for (C) (left to right): <0.0001, N/D, <0.0001, 0.4002, <0.0001, 0.7071, <0.0001, 0.1288, <0.0001 and 0.7984. (D) Scatterplot of the genome-wide average fold change (FC) of ribosome and mRNA abundance in RBM15B-depleted MOLM-13 (*MLL-AF9*) cells (Ribo-seq, log2FC > 0.15 and <−0.15, FDR <0.05; RNA-seq, log2FC >0.5 and <−0.5, FDR <0.05). (E) Genome Browser views of mRNA (RNA-seq) and ribosome-protected fragments (ribo-seq) signals for *MYC, HOXA9,* and *BCL2* with downregulated TE upon RBM15B knockdown. (F, G) Polysome profiles and polysome/sub-polysome ratio changes of *MYC, HOXA9,* and *BCL2* upon RBM15B knockdown (F) and EPZ5676 treatment (G) in MOLM-13 (*MLL-AF9*) cells. *GAPDH* mRNA acted as the negative control. The *P* values for (F) (left to right): 0.0150, 0.0008, 0.0023, and 0.1824; The *P* values for (G) (left to right): 0.0008, 0.0078, <0.0001, and 0.5958. (H) Western blot for MYC, HOXA9, and BCL2 in the RBM15B knockdown in MOLM-13 (*MLL-AF9*) cells. (I) Western blot for MYC, HOXA9, and BCL2 in the DOX-induced overexpression in MOLM-13 (*MLL-AF9*) cells. (J) Western blot for MYC, HOXA9, and BCL2 in *MLL*-r or *MLL*-wt leukemia patient primary cells electrotransfected with si-NC or si-RBM15B. AML acute myelocytic leukemia, ALL acute lymphocytic leukemia. *MLL-AF9* and *MLL-AF10* indicate *MLL*-fusion genes. *MLL*-wt indicates *MLL* wild-type. (K) Schematic for GFP reporter system including TWINS^m6Awt-GFP and TWINS^m6Amut-GFP. The constructs: one is TWINS^m6Awt-GFP (DNA sequences containing the native promoter and TWINS region with m6A sites from *HOXA9* or *MYC* fused to the GFP sequence), and another one is TWINS^m6Amut-GFP (the native promoter and TWINS region with m6A mutation sites from *HOXA9* or *MYC* fused to the GFP sequence). (L, M) GFP-MFI/RE-RNAs were analyzed by flow cytometry and qRT-PCR. Upregulation or downregulation of RBM15B via transfecting CMV-RBM15B plasmid (L) or si-RBM15B (M) into TWINS^m6Awt-GFP or TWINS^m6Amut-GFP constructed cells, and regulate the GFP translation efficiency. GFP-MFI/RE-RNAs: GFP-MFI/relative expression of mRNAs. *P* values for (L): *P* < 0.0001 (TWINS^m6Awt + NC vs TWINS^m6Amut + NC), *P* = 0.0004 (TWINS^m6Awt + NC vs TWINS^m6Awt + RBM15B), and *P* = 0.1048 (TWINS^m6Amut + NC vs TWINS^m6Amut + RBM15B). *P* values for (M): *P* < 0.0001 (TWINS^m6Awt + si-NC vs TWINS^m6Amut + si-NC), *P* = 0.0005 (TWINS^m6Awt + si-NC vs TWINS^m6Awt + si-RBM15B), and *P* = 0.7090 (TWINS^m6Amut + si-NC vs TWINS^m6Amut + si-RBM15B). Values are mean ± SEM (*n* = 3), one-way ANOVA in (L, M), and unpaired two-tailed Student's *t* test in (B, C, F, G). NS not significant; * *P* < 0.05; **P* < 0.01; ***P* < 0.001. Source data are available online for this figure.

TWINS^m6Awt-GFP and TWINS^m6Amut-GFP cells (Appendix Fig. S11). GFP-MFI/RE-RNAs results showed that the EPZ5676 treatment dramatically reduced the GFP translation efficiency both in TWINS^m6Awt-GFP of HOXA9 and TWINS^m6Awt-GFP of MYC cells. No significant effect was observed in the mutant reporters. These findings suggest that H3K79me2 not only regulates target gene transcription but also enhances the translation of oncogenic transcripts by facilitating RBM15B-mediated selective m6A site modification. Altogether, these results indicate that the uniquely selective TWINS m6A modification mediated by the H3K79me2–RBM15B axis can specifically increase target protein levels by facilitating translation efficiency, thereby driving tumorigenesis.

## Targeting H3K79me2-RBM15B-m6A axis impairs leukemia progression in vivo

Previous studies have shown that selective m6A modifications of oncogenic RNAs could confer tumorigenesis. To investigate whether targeting H3K79me2-RBM15B-m6A axis impairs leukemia progression, we employed three different mice models, including the xenograft model using MOLM-13 cells, *MLL-AF9* leukemia mice model and patient-derived xenograft (PDX) mouse models (Fig.6; Appendix Figs. S12–14). Strikingly, all of these mice models have showed that suppression of H3K79me2-RBM15B-m6A axis by targeting RBM15B (Rbm15b in murine) led to decreased infiltration ability for *MLL*-r leukemic cells and less splenomegaly and liver injury; while, it has no effect in the PDX of *MLL*-wt-driven leukemia (Fig. 6A–F; Appendix Figs. S12B–D, 13B, and 14B–D). The differentiation block characteristic of leukemia cells was also significantly relieved upon RBM15B reduction in organs (Appendix Fig. S12E–H), implying that RBM15B is necessary to inhibit leukemic cell differentiation in vivo. Consistent with these results, mice in the sh-RBM15B *MLL*-r leukemia groups survived longer than did those in the sh-NC groups (Fig. 6G,H; Appendix Figs. S12J and 14E), suggesting that suppression of H3K79me2-RBM15B-m6A axis could inhibit *MLL*-r leukemia progression.

Cancer stem cell confers the inevitable recurrence of tumors, targeting of which is an effective strategy for cancer treatment, but still face various challenges (Yang et al, 2020). We further examined whether blockade of H3K79me2-RBM15B-m6A axis impairs LSC maintenance. As expected, downregulation of Rbm15b decreased the percentage of GFP+Lin-c-Kit+Sca-1+ (LSK) and L-GMP (LSK Granulocyte-macrophage progenitors, GFP+Lin-c-Kit+Sca-1 + CD34 + CD16/32 + ) cells in the BM of *MLL-AF9*-driven leukemia mice (Fig. 6I; Appendix Fig. S14F). In addition, mice harboring a sh-RBM15B PDX of *MLL-AF4*-driven leukemia had a low infiltration ability, compromised tumor burden, alleviated differentiation block, decreased population of human CD45 + CD34 + CD38- cells, and had a longer survival; while no effects were found in the sh-RBM15B PDX of *MLL*-wt-driven leukemia (Fig. 6J; Appendix Fig. S14G,H). In brief, these in vivo studies suggest that RBM15B is required for leukemia maintenance, which it may facilitate by promoting LSC self-renewal, cell survival, and differentiation blockage.

We further determine whether H3K79me2-RBM15B-m6A axis could be therapeutically targeted. IC50 assays showed that compared to *MLL*-wt leukemia cells, *MLL*-r leukemia cells were significantly more sensitive to STM2457 (a METTL3 inhibitor) but not to FB23-2 (a FTO inhibitor) (Appendix Fig. S15A,B), supporting that m6A modifications mediated by RBM15B/METTL3 complex have important functions in *MLL*-r leukemia. Moreover, the CCK-8 assays showed that the combination of STM2457 and EPZ5676 compounds can better suppress the *MLL*-r leukemia cell viability than STM2457 or EPZ5676 treatment alone (Appendix Fig. S15C,D). These data suggested that inhibition of the H3K79me-RBM15B/METTL3 complex-m6A axis can dramatically restrain *MLL*-r leukemia.

We further assessed the effects of combining RBM15B suppression and H3K79me2 suppression, which is double-targeting this epigenetic–epitranscriptomic axis modification. As expected, cell proliferation was significantly more inhibited by combination treatment than by control or mono-treatment (Fig. 7A). Of note, we found that both RBM15B protein level and

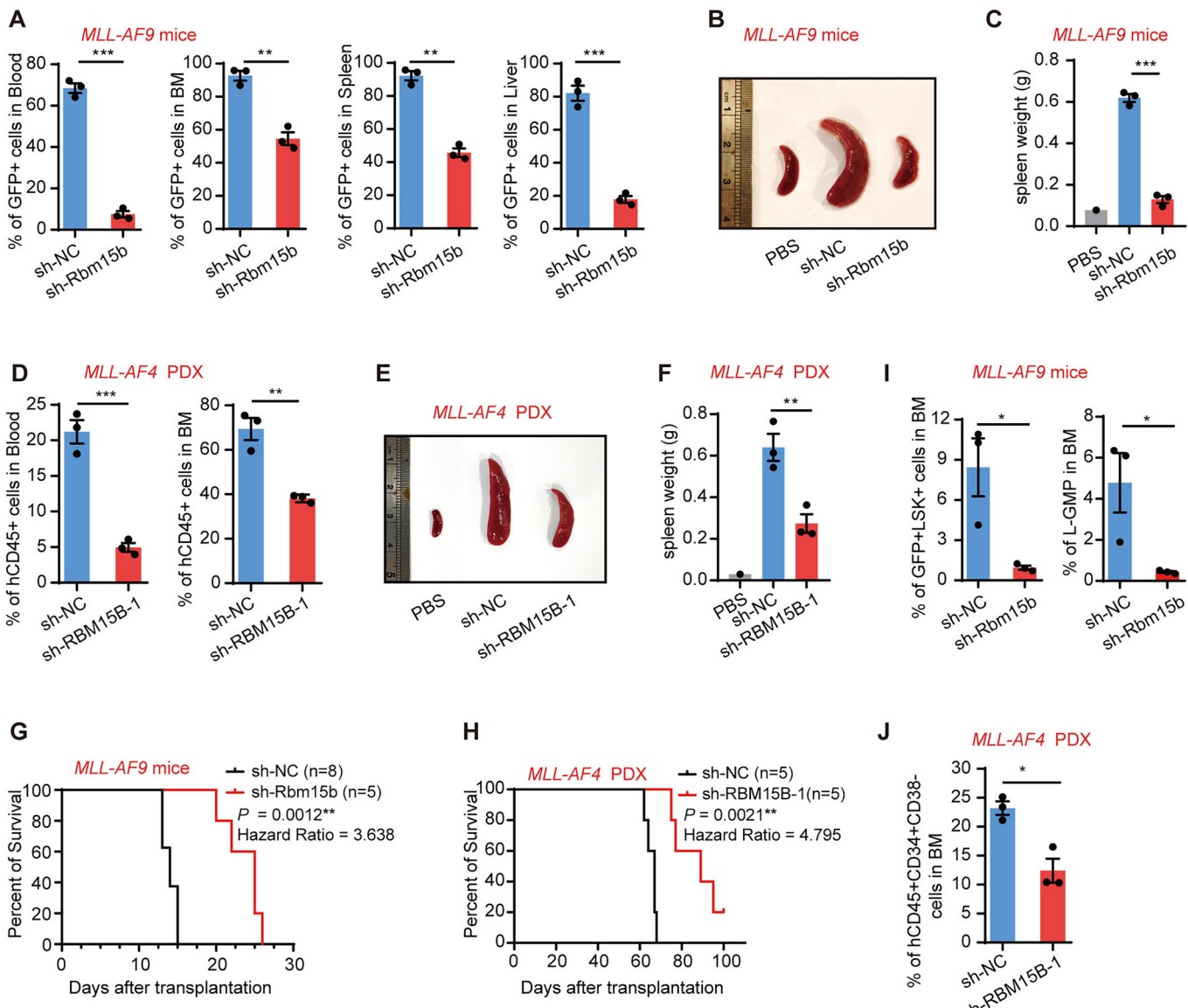

**Figure 6.    Targeting H3K79me2-RBM15B axis impairs *MLL*-r leukemia progression in vivo.**

(**A**) *MLL-AF9*-driven mice leukemia cells in recipient mice organs (BM, blood, spleen, liver) were analyzed by flow cytometry. The P values (left to right): <0.0001, 0.0014, 0.0028, and 0.0002. (**B, C**) Size (**B**) and weight (**C**) of the spleen of the recipient mice transplanted with sh-Rbm15b or sh-NC *MLL-AF9* leukemia cells. Mice treated with PBS were used as the negative controls. P < 0.0001. (**D**) Flow cytometry quantitative results for human CD45⁺ cells in blood (P = 0.0008) and bone marrow (P = 0.0035) of mouse recipients of sh-NC or sh-RB15B infected *MLL-AF4* patient primary cells. (**E, F**) Size (**E**) and weight (**F**) of the spleen of the recipient mice transplanted with sh-RBM15B-1 or sh-NC *MLL-AF4* primary cells (8 weeks after transplantation). Mice treated with PBS were used as the negative controls. P = 0.0099. (**G**) Kaplan–Meier survival curves of recipient mice transplanted with *MLL-AF9* driven mice leukemia cells after KD of Rbm15b. The P values and hazard ratio were calculated by log-rank test. (**H**) Kaplan–Meier plot showing the survival time of recipient mice transplanted with patient primary *MLL-AF4* leukemia cells. The P values and hazard ratio were calculated by log-rank test. (**I**) Flow cytometry quantitative analysis of GFP + LSK+ cells (left, P = 0.0259) and L-GMP cells (right, P = 0.0397) in the BM of recipient mice transplanted with *MLL-AF9* leukemia cells after knockdown Rbm15b. (**J**) Proportion of leukemia stem cells (CD34⁺CD38⁻) isolated from *MLL-AF4* mice bone marrow. P = 0.0101. Values are mean ± SEM (n = 3), one-way ANOVA in (**C**), and unpaired two-tailed Student's t test in (**A, D, F, I, J**). NS, not significant; * P < 0.05; **P < 0.01; ***P < 0.001. Source data are available online for this figure.

TWINS m⁶A levels of *MLL*-r targets were lower in healthy CD34+ cells (Appendix Fig. S16A,B). Double-targeting RBM15B and H3K79me2 had a minimal effect on the differentiation and clone formation of healthy CD34+ cells, suggesting the impact of RBM15B on normal hematopoiesis could be little (Appendix Fig. S16C–E). Strikingly, combining RBM15B suppression and

H3K79me2 suppression markedly downregulated the protein levels of MYC, HOXA9 and BCL2 in leukemia cells (Fig. 7B,C); but the impact on normal hematopoiesis could be little (Appendix Fig. S16F), suggesting the drug combination might more strongly inhibit the oncoprotein level of target genes regulated by H3K79me2–RBM15B–m⁶A axis in leukemia. In vivo studies further

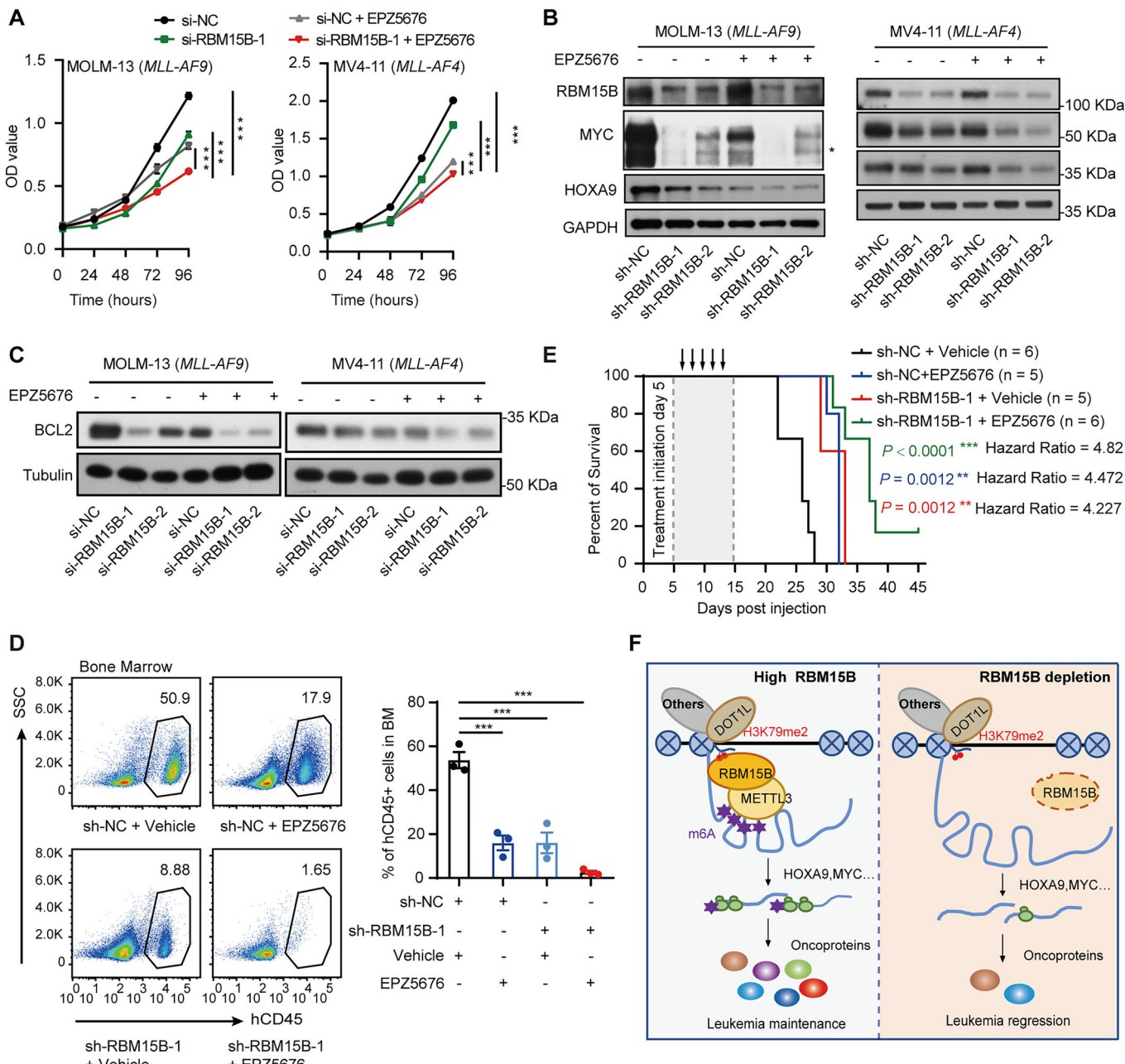

**Figure 7. Double-targeting H3K79me2-RBM15B-m⁶A axis dramatically compromised tumor burden and improved survival.**

(**A**) Proliferation of MOLM-13 and MV4-11 cells in four groups: si-NC (empty vector control), si-RBM15B (RBM15B knockdown), si-NC + EPZ5676 (EPZ5676 treatment), and si-RBM15B + EPZ5676. Data are means ± SEM of three biological replicates and were analyzed by two-way ANOVA (**$P < 0.01$, ***$P < 0.001$). $P$ values for MOLM-13: $P < 0.0001$ (si-NC vs si-RBM15B-1 + EPZ5676), $P < 0.0001$ (si-RBM15B-1 vs si-RBM15B-1 + EPZ5676) and $P < 0.0001$ (si-NC + EPZ5676 vs si-RBM15B-1 + EPZ5676). $P$ values for MV4-11: $P < 0.0001$ (si-NC vs si-RBM15B-1 + EPZ5676), $P < 0.0001$ (si-RBM15B-1 vs si-RBM15B-1 + EPZ5676), and $P < 0.0003$ (si-NC + EPZ5676 vs si-RBM15B-1 + EPZ5676). (**B, C**) Western blot for MYC, HOXA9, and BCL2 in the RBM15B knockdown in MOLM-13 and MV4-11 cells treated with or without EPZ5676. GAPDH acted as the internal control. (**D**) Representative image and histogram plots showing the human CD45+ levels in BM derived from the mice after implantation. Data are means ± SEM of three biological replicates and were analyzed by one-way ANOVA (***$P < 0.001$). All of the $P$ values: $P < 0.0001$. (**E**) Kaplan–Meier survival curves of mice injected with sh-NC or sh-RBM15B MOLM-13 cells after vehicle and EPZ5676 treatment at day 5. Vehicle or 50 mg/kg EPZ5676 was administered every other day by intraperitoneal injection, for a total of five treatments. $P$ values and hazard ratios were calculated using the log-rank test (**$P < 0.01$, ***$P < 0.001$). $P < 0.0001$ (sh-NC vs sh-RBM15B-1 + EPZ5676), $P = 0.0012$ (sh-NC vs sh-RBM15B-1), and $P = 0.0012$ (sh-NC vs EPZ5676). (**F**) A working model proposed for RBM15B-guided, specifically selective m⁶A modification by directly binding H3K79me2 and recruiting the MTC. Source data are available online for this figure.

exhibited that combining RBM15B knockdown and H3K79me2 suppression dramatically compromised tumor burden and improved survival (Fig. 7D,E), which may suggest the potential strategy of suppression combination for *MLL*-rearranged leukemia treatment exclusively.

Altogether, our observations reveal that the uniquely selective TWINS m⁶A modification mediated by the H3K79me2–RBM15B axis can specifically increase target protein levels by facilitating translation efficiency, thereby promoting leukemia progression (Fig. 7F). Suppression of tumorigenesis through RBM15B depletion is dictated, at least partially, by disturbing the H3K79me2–RBM15B–m⁶A axis.

## Discussion

Dysregulation of m⁶A modification has been acknowledged as an epigenetic feature of various diseases. With the development of single-base m⁶A methylation sequencing, several studies have exhibited that m⁶A modification have RNA substrate and locus selectivity and may administrate diverse RNA fate (Barbieri et al, 2017; Einstein et al, 2021; Hu et al, 2022; Huang et al, 2019a; Patil et al, 2016; Zaccara et al, 2019). However, the cause of site-selectivity of m⁶A modification on RNA substrate and its role in tumorigenesis have not been well explored. In this study, we established that RBM15B, which is more highly expressed in *MLL*-r leukemias, could serve as a critical risk factor. RBM15B recognizes H3K79me2 and mainly regulates m⁶A modification in 5′ untranslated regions and around the start codons (TWINS region) of mRNA transcribed from target genes of MLL-fusion proteins. We further illustrated that RBM15B, identified as an epigenetic transductor, mediates TWINS region m⁶A deposition mainly through its H47 residue, which directly binds to H3K79me2, the most important epigenetic regulator in *MLL*-r leukemia. The selective m⁶A modification orchestrated by H3K79me2–RBM15B axis could facilitate translation efficiency and the progression of *MLL*-r leukemia. This regulatory paradigm first explained the site-selectivity of m⁶A modification in hematopoietic malignancy, and also suggests that H3K79me2 has a posttranscriptional regulation function. Our work also reports a previously unknown crosstalk between H3K79me2 and m⁶A methylation.

RBM15B is a well-known RNA-binding protein (Shi et al, 2019). It plays a role in m⁶A MTC members and recruiting the m⁶A MTC to specific RNA regions by recognizing RNA motifs (Huang et al, 2021). For example, X-inactive specific transcript (XIST)-mediated gene silencing requires m⁶A modification, which depends on the binding of RBM15B to the regions adjacent to the 5′ end of XIST (Patil et al, 2016). However, the exact regulatory mechanism behind this effect remains unknown. In this study, we revealed a novel mechanism in which RBM15B guides specific m⁶A deposition. We demonstrated that RBM15B can choose target RNAs and precise m⁶A modification sites through a direct epigenetic transduction from H3K79me2 to transcribed RNAs. This is a novel function of RBM15B that can select both target transcripts and modified sites on RNAs through histone modification recognition, especially in an *MLL*-rearranged context. We also noticed that H3K4me3 has a weak interact with RBM15B, thus, we cannot rule out that whether some other histone modifications may assist to the process of H3K79me2-RBM15B-guided m⁶A-selective deposition in TWINS

region which deserve to be further declared. In addition, RBM15B functions as a cofactor to regulated mRNA splicing and export (Loyer et al, 2011; Zolotukhin et al, 2009). Given the multifaceted roles of RBM15B in regulating co-transcription and posttranscription, RBM15B may act in distinct regulation manners and its higher expression in *MLL*-rearranged contexts might be regulated by many factors, for example, MLL-fusion protein. Furthermore, we also observed that knockdown of RBM15B causes an increase in m⁶A signal density in the distal end of 3′ UTRs. Previous studies showed that 3′ UTR m⁶A can promote or inhibit translation, and the effect may be dependent on the modified specific location, cell environment, and the protein type (Corovic et al, 2025; Meyer, 2019). This suggests that the RBM15B-midated relocation of m⁶A on different transcripts may have different roles in translation. Thus, it will be important to further investigate the underlying mechanisms of up/downstream in the RBM15B-regulated pathway.

How m⁶A is precisely deposited at proper sites within the transcriptome to affect transcript fate remains poorly understood. m⁶A modification in the CDS and 3′ UTR of nascent RNAs via crosstalk guided by H3K36me3 indicates that selective m⁶A modification exists among multiple potential m⁶A consensus sequences (Huang et al, 2019a). However, partial m⁶A modification peaks in the 5′ UTR, or the mechanism regulating the selectivity of m⁶A deposition is still unclear. Our study demonstrated selective and precise m⁶A deposition in the TWINS region of RNAs, particularly those from target genes of MLL-fusion proteins, via crosstalk directly guided by H3K79me2 modification. We revealed the mechanism of specific, selective, and precise m⁶A modification in the TWINS region of RNA and elucidated the mechanism of precise crosstalk between epigenetic and epitranscriptomic modifications. Interestingly, a recent study found that EJCs colocalize the splice sites and can inhibit the m⁶A modification of exon junction–proximal RNA within coding sequences but not long internal exons and terminal exons, which are notably free of EJCs (He et al, 2023). These data together underlie multiple characteristics of RNA m⁶A modification specificity, which may be orchestrated, at least partially, by distinct regulatory complexes in various contexts.

The regulators of m⁶A modification are regarded as potential candidates to influence m⁶A modification to affect leukemia progression (Li et al, 2017b; Vu et al, 2019; Weng et al, 2022; Weng et al, 2018). For example, METTL14 is required for AML development, and its silencing inhibits AML by interfering with m⁶A modification of the oncogenes(Weng et al, 2018). FTO enhances leukemic oncogene-mediated cell transformation (Li et al, 2017b). However, these core m⁶A regulators have been reported to play important roles in embryonic development and normal cells (Wang et al, 2014; Wei et al, 2022; Xu et al, 2021). Factors with greater specificity in m⁶A selective modification in certain contexts would be better targets with fewer side effects. We found that RBM15B knockdown specifically inhibits the progression of *MLL*-r leukemia but not of *MLL*-wt ones. So, RBM15B might be a more promising therapeutic target for certain leukemias with *MLL*-rearrangement because its downregulation might have only a modest effect on healthy hematopoietic cells. In addition, inhibiting both H3K79 modification and H3K79me2–m⁶A crosstalk by silencing RBM15B can greatly suppress the progression of *MLL*-r leukemia. This finding suggests that double-targeting the epigenetic–epitranscriptomic modification axis might offer a novel

effective therapeutic strategy for aggressive hematopoietic malignancy.

In summary, we demonstrated that RBM15B, identified as an epigenetic transducer, can specifically regulate the uniquely selective m$^6$A modification in the TWINS region of RNA by recognizing H3K79me2. The special H3K79me2–RBM15B-m$^6$A axis could control translation efficiency of oncogenic transcripts.

Moreover, we established that highly RBM15B is a risk factor and has a pro-neoplastic effect in *MLL*-r leukemias. Inhibiting H3K79me2-RBM15B-m$^6$A axis can significantly block *MLL*-r leukemia progression. Our findings not only partly explained the selective model of m$^6$A deposition in the TWINS region of RNAs but also suggest that double-targeting the epigenetic–epitranscriptomic modification axis might offer a novel therapeutic strategy for aggressive hematopoietic malignancy. It is also the first report about the crosstalk between H3K79me2 and m$^6$A methylation.

# Methods

### Reagents and tools table

| Reagent/resource | Reference or source | Identifier or catalog number |
| --- | --- | --- |
| **Experimental models** | | |
| MOLM-13 | DSMZ | Cat# ACC 554; RRID:CVCL_2119 |
| MV4-11 | ATCC | Cat# CRL-9591;RRID:CVCL_0064 |
| THP1 | ATCC | Cat# TIB-202; RRID:CVCL_0006 |
| RS4;11 | ATCC | Cat# CRL-1873; RRID:CVCL_0093 |
| HL60 | ATCC | Cat# CCL-240; RRID:CVCL_0002 |
| NB4 | DSMZ | Cat# ACC 207; RRID:CVCL_0005 |
| Jurkat | ATCC | Cat# CRL-2899; RRID:CVCL_U617 |
| K562 | ATCC | Cat# CCL-243; RRID:CVCL_0004 |
| HEK293T | ATCC | Cat# CRL-1573; RRID:CVCL_0045 |
| Human primary patient samples | This paper | N/A |
| NOD.CB17-*Prkdc*$^{scid}$/NcrCrl mice | Charles River Laboratories | Cat# 406 |
| Xenograft AML mouse model | This paper | N/A |
| **Recombinant DNA** | | |
| pCDH-CMV-MCS-EF1-Puro | SYSTEM BIOSCIENCES | Cat# CD510B-1 |
| CMV-RBM15B-FLAG | This paper | N/A |
| CMV-RBM15B-△493-891 | This paper | N/A |
| CMV-RBM15B-△1-699 | This paper | N/A |
| CMV-RBM15B-△RRM1/2/3 | This paper | N/A |

| Reagent/resource | Reference or source | Identifier or catalog number |
| --- | --- | --- |
| CMV-RBM15B-△219-492 | This paper | N/A |
| CMV-RBM15B-△1-219 | This paper | N/A |
| CMV-RBM15B-△K79HMBD | This paper | N/A |
| CMV-RBM15B$^{H47A}$ | This paper | N/A |
| CMV-RBM15B-HA | This paper | N/A |
| HOXA9-TWINS$^{m6Awt}$-GFP | This paper | N/A |
| HOXA9-TWINS$^{m6Amut}$-GFP | This paper | N/A |
| MYC-TWINS$^{m6Awt}$-GFP | This paper | N/A |
| MYC-TWINS$^{m6Amut}$-GFP | This paper | N/A |
| pCW-Cas9 | Addgene | Cat# 50661 |
| pCW-RBM15B-HA | This paper | N/A |
| pET-N-GST-Thrombin-C-His | Beyotime Biotechnology | Cat# D2911 |
| pET-GST-RBM15B | This paper | N/A |
| pGreenPuro-shRNA | SYSTEM BIOSCIENCES | Cat# SI505A-1 |
| pGreen-sh-RBM15B | This paper | N/A |
| pGreen-sh-Rbm15b | This paper | N/A |
| **Antibodies** | | |
| Rabbit anti-RBM15B | Proteintech | 22249-1-AP |
| Rabbit anti-METTL3 | Proteintech | 15073-1-AP |
| Rabbit anti-METTL14 | Novus Biologicals | NBP1-81392 |
| Mouse anti-WTAP | Proteintech | 60188-1-Ig |
| Rabbit anti-FTO | Cell Signaling Technology | 31687 |
| Rabbit anti-DOT1L | Cell Signaling Technology | 77087 |
| Rabbit anti-MLL1 | Cell Signaling Technology | 14689 |
| Rabbit anti-BRD4 | Cell Signaling Technology | 13440 |
| Rabbit anti-H3K36me3 | Abcam | ab9050 |
| Rabbit anti-H3K79me2 | Abcam | ab3594 |
| Rabbit anti-H3K4me3 | Abcam | ab8580 |
| Rabbit anti-H3 | Abcam | ab1791 |
| Rabbit anti-H4 | Abcam | ab177840 |
| Rabbit anti-C-MYC | Cell Signaling Technology | 18583 |
| Rabbit anti-HOXA9 | Proteintech | 18501-1-AP |
| Rabbit anti-MEN1 | Proteintech | 15159-1-AP |
| Rabbit anti-GAPDH | Proteintech | 10494-1-AP |
| Rabbit anti-β-Tubulin | Cell Signaling Technology | 2146S |
| Rabbit anti-m$^6$A | Synaptic Systems | 202 003 |
| Mouse anti-m$^5$C | Abcam | Ab10805 |
| Rabbit anti-IgG | Millipore | 12-370 |

| Reagent/resource | Reference or source | Identifier or catalog number |
|---|---|---|
| Rabbit anti-GST | Cell Signaling Technology | 2622 |
| Mouse anti-FLAG | Sigma-Aldrich | F1804 |
| Mouse anti-hCD11b-APC | BD Biosciences | 550019 |
| Mouse anti-hCD14-PE | Thermo Fisher Scientific | 12-0149-42 |
| Mouse anti-hCD14-APC | BD Biosciences | 555399 |
| Mouse anti-hCD19-APC | BD Biosciences | 7108618 |
| Mouse anti-hCD45-FITC | Thermo Fisher Scientific | 11-0459-42 |
| Anti-hCD34-PE | Miltenyi Biotec (MACS) | 130-132-790 |
| Mouse anti-hCD38-BV421 | BD Biosciences | 4026292 |
| Anti-Mouse Hematopoietic Lineage Antibody Cocktail-eFluor 450 | eBioscience | 88-7772-72 |
| Anti-Mo Ly-6A/E(Sca-1)-APC | eBioscience | 17-5981-81 |
| Cyanine7 anti-mouse CD117(c-kit)-APC | Biolegend | 135136 |
| Anti-Mouse CD34-PE | BD Biosciences | 551387 |
| Anti-Mouse CD16/32-MAB-PE | Elabscience | E-AB-F0997UI |
| Anti-Mo CD11b-APC | eBioscience | 17-0112-81 |
| Anti-Mo Ly-6G/Ly-6C-PE | eBioscience | 12-5931-81 |
| AlexaFluor 488-conjugated secondary antibodies(mouse) | Thermo Fisher Scientific | A28175 |
| AlexaFluor 594-conjugated secondary antibodies (rabbit) | Thermo Fisher Scientific | A-21207 |
| Mouse HRP-conjugated secondary antibody | Cell Signaling Technology | 7076 |
| Rabbit HRP-conjugated secondary antibody | Thermo Fisher Scientific | 31460 |
| **Oligonucleotides and other sequence-based reagents** | | |
| Primers | This Paper | See Appendix Table S4 |
| siRNA | This Paper | See Appendix Table S5 |
| shRNA | This Paper | See Appendix Table S5 |
| **Chemicals, enzymes, and other reagents** | | |
| StemSpan SFEM II | Stemcell | 09605 |
| Recombinant Human IL-6 | PeproTech | 200-06 |
| Recombinant Human IL-3 | PeproTech | 200-03 |
| Recombinant Human Flt3-Ligand | PeproTech | 300-19 |
| Recombinant Human SCF | PeproTech | 300-07 |
| Recombinant Human TPO | PeproTech | 300-18 |
| EPZ5676 | Selleck | S7062 |
| iBET151 | Selleck | S2780 |

| Reagent/resource | Reference or source | Identifier or catalog number |
|---|---|---|
| STM2457 | Selleck | S9870 |
| FB23-2 | Selleck | S8837 |
| Puromycin | Beyotime Biotechnology | ST551 |
| Nocodazole | Sigma-Aldrich | M1404 |
| Cycloheximide (CHX) | Sigma-Aldrich | 239764 |
| Actinomycin D | MedChemExpress | HY-17559 |
| Doxycycline | Selleck | S4163 |
| Dimethyl sulfoxide (DMSO) | Sigma-Aldrich | D2650 |
| IPTG | TransGen Biotech | GF101-01 |
| Recombinant Histone H3K79me2 | Active Motif | 31599 |
| Recombinant Histone H3.3 | Active Motif | 31295 |
| **Software** | | |
| FlowJo | FLOWJO, LLC | https://www.flowjo.com/solutions/flowjo; RRID: SCR_008520 |
| Fiji | Schindelin et al, 2012 | https://imagej.net/software/fiji/ |
| GraphPad Prism 8 | GraphPad Software | https://www.graphpad.com/scientificsoftware/prism/; RRID:SCR_002798 |
| Bowtie2 (v2.5.0) | Langmead and Salzberg, 2012 | |
| Hisat2 (v2.2.1) | Kim et al, 2019 | |
| SAMtools (v1.3.1) | Danecek et al, 2021 | |
| exomePeak2 (v1.6.1) | Meng et al, 2013 | |
| metaplotR | Olarerin-George and Jaffrey, 2017 | |
| MACS2 (v2.2.9.1) | Zhang et al, 2008 | |
| GLORI-tools pipeline | Shen et al, 2024 | |
| Cutadapt (v4.4) | Kechin et al, 2017 | |
| Seqkit (v2.5.1) | Shen et al, 2016 | |
| Deeptools (v3.5.4) | Ramírez et al, 2016 | |
| Bedtools intersectBed (v2.31.0) | Quinlan, 2014 | |
| StringTie (v2.1.7) | Kovaka et al, 2019 | |
| DESeq2 (v1.38.3) | Love et al, 2014 | |
| Riborex (v2.4.0) | Li et al, 2017a | |
| **Other** | | |
| Neon Transfection System | Invitrogen | Cat# MPK1096 |
| Multi SIM X | NanoInsights-Tech | N/A |
| BD FACSCelesta | BD Biosciences | N/A |

## Cell lines and patient samples

MOLM-13, THP1, RS4;11, NB4, HL60, Jurkat, and K562 cells are cultured in RPMI 1640 (Hyclone) supplemented with 10% FBS (Hyclone). MV4-11 and 293 T is cultured in IMDM (Hyclone) and DMEM (Gibco) supplemented with 10% FBS, respectively. Cells were kept at 37 °C and 5% $CO_2$. Primary cells derived from *MLL*-r leukemia patients and healthy individuals were either cultured in SFEM medium supplemented with 20% FBS and 10 ng/mL of SCF, TPO, Flt-3 L, IL-3, and IL-6 (Weng et al, 2018). All clinical samples and CD34+ cells from healthy individuals were obtained with informed consent from the First Affiliated Hospital of Sun Yat-sen University and approved by the Hospital's Protection of Human Subjects Committee with informed consent from patients (approved No. [2023]249). The detailed clinicopathological characteristics of the patients were summarized in Appendix Tables S1–3. All cell lines were recently tested; no mycoplasma contamination was detected.

## RNA isolation and quantitative real-time PCR (qRT-PCR)

RNA was extracted from cell samples by using TRIzol reagent (Invitrogen, USA) according to the manufacturer's instructions. The RNA was reverse-transcribed using RT reagent Kit RR047A (Takara, Japan), and following quantitative PCR (qPCR) analysis was performed using TB Green premix ExTaq Real-Time PCR kit (Takara, Japan) on an ABI QuantStudio 7 Flex PCR system (Applied Biosystems, USA) with primers listed in Appendix Table S4. Relative expression levels were calculated using the 2^-ΔΔCT method.

## Plasmid construction and RNA interference

Constitutive gene expression was performed by cloning the cDNA of interest into pCDH-CMV-MCS-EF1-Puro-copGFP (pCDH) vectors (Addgene, USA). Doxycycline (Dox)-inducible gene expression was performed by cloning gene of interest into pCW vector, which was modified by deleting the Cas9 sequences from the original pCW-Cas9 plasmid (Addgene, USA). For protein purification, the cDNAs of RBM15B were cloned into pET-N-GST-Thrombin-C-His vector (Beyotime Biotechnology) to express GST-RBM15B-HIS proteins. All constructs were verified by DNA sequencing. Stable RNA interference was performed using shRNA, which was cloned into pGreenPuro shRNA cloning and Expression Vector (System Biosciences). Transient RNA interference was performed using siRNA (Genepharma). The siRNA and shRNA sequences were listed in Appendix Table S5.

The HOXA9/MYC-TWINS^m6Awt/mut-GFP reporter system plasmids were cloned into pCDH-CMV-MCS-EF1-Puro-copGFP (pCDH) vectors, which CMV promoter was excised and replaced by the native promoter from *HOXA9* or *MYC*. $m^6A$ site in HOXA9/MYC TWINS region were identified by GLORI-seq. The mutated $m^6A$ site (RRACH sites) was achieved through codon substitutions, with the principle of maintaining comparable translation efficiency. The primers were listed in Appendix Table S4.

## Cell transfection and lentivector expression systems

Transient transfections of an overexpression vector in adherent cells were performed using Lipofectamine 2000 reagent (Invitrogen) according to the manufacturer's instructions. Transient

electrotransfection of siRNAs in suspension cells was performed using Neon Transfection System (Invitrogen) according to the manufacturer's guidelines. For lentivirus infection to generate stable cell lines, pGreenPuro vector for knockdown with shRNA and pCDH-CMV-MCS-EF1-Puro-copGFP (pCDH) vector for constitutive gene over-expression were used for virus packaging with Lentivector Expression Systems (System Biosciences, Germany), which consisted of pPACKH1-GAG, pPACKH1-REV, and pVSV-G. For Dox-inducible gene expression, pCW (Tet-on promoter), pMD2.G, and psPAX2 (Addgene) vectors were used for lentivirus particle production. The lentivirus-infected cells were selected with puromycin.

## Protein extraction

Total protein was extracted from cell samples using RIPA lysis buffer (Beyotime Biotechnology) supplemented with 1×complete ULTRA protease inhibitor (Roche). TRIzol reagent (Invitrogen) was used for protein extraction from patient samples according to the manufacturer's instructions.

## Immunoprecipitation (IP)/co-immunoprecipitation (co-IP)

Cell extraction was performed using NP40 buffer (150 mM NaCl, 1.5 mM $MgCl_2$, 0.5% NP40, 50 mM Tris-HCl at pH 8.0) supplemented with Thermo Scientific™ Halt™ Protease Inhibitor Cocktail (Thermo Fisher Scientific). After sonication, about 1000 μg of cell lysates were used for immunoprecipitation with 2 μg primary antibody overnight at 4 °C. The next day, the antibody-bound protein of interest in NP40 buffer was incubated with 40 μl of MagnaBind Protein A/G Beads. After three washes with Wash Buffer (10 mM Tris-HCl pH 7.5, 1 mM EDTA, 1 mM EGTA, 150 mM NaCl, 1% Triton-X, 0.2 mM sodium orthovanadate), protein-bound beads were mixed with 5× loading buffer (Fdbio scienceto) the final concentration of 1× loading buffer and boiled for 10 min at 95 °C. The samples were then stored at −20 °C or ready for SDS-PAGE (sodium dodecyl sulfate–polyacrylamide gel electrophoresis). To explore whether protein interaction was mediated by RNA or DNA, RNase A (Takara) or DNase I was added to the cell lysate to a final concentration of 20 μg/ml or 100U/ml and incubated for 30 min at 37 °C before IP procedures.

## Western blot analysis

Protein extracts were resolved by SDS-PAGE and then electro-phoretically transferred onto a PVDF membrane (EMD Millipore). The membrane was incubated for 1 h in blocking buffer (1×TBST containing 5% BSA), and then incubated at 4 °C overnight with the primary antibody. After three washes with 1×TBST, the membrane was incubated for 1 h with horseradish peroxidase (HPR)-conjugated secondary antibody (EMD Millipore). Membrane was visualized using the Immobilon Western Chemiluminescent HRP Substrate (EMD Millipore). The densitometric ratio of protein bands was calculated by ImageJ.

## Cell proliferation assay and flow cytometric analysis

Cell proliferation assays were performed using Cell Counting Kit-8 (Dojindo Molecular Technologies). Briefly, cells were seeded in a

96-well plate with 100 μl of complete medium per well. CCK-8 reagent was added to each well 4 h prior to measurement. Absorbance at 480 nm and 630 nm was measured by a VICTORTM X5 Multilabel Plate Reader (PerkinElmer) at the indicated time points.

For cell cycle analysis, cells were synchronized by treating with 30 ng/ml nocodazole for 24 h before analysis. Cell samples were then harvested and incubated with propidium iodide solution (Lianke) in the dark for 30 min at room temperature. For cell differentiation analysis, cells were incubated with human APC-conjugated CD11b (BD Biosciences) and PE-conjugated CD14 (BD Biosciences) antibodies. All the samples for flow cytometric analysis were performed on a BD FACSCelesta (BD Biosciences, USA).

## GFP-MFI/RE-RNAs calculating

The HOXA9/MYC-TWINS^m6Awt/mut^-GFP reporter cells were first treated with RBM15B overexpression/knockdown and EPZ5676 treatment. GFP translational output was quantified by measuring the mean fluorescence intensity (MFI) via flow cytometry. Concurrently, the relative expression levels of GFP mRNA were detected by qRT-PCR. Finally, the GFP-MFI values were normalized to the corresponding mRNA levels (GFP-MFI/relative mRNA expression).

## Methylcellulose colony assays

A total of 1000 cells were used per colony-forming cell assay and were added to 1 ml of methylcellulose media (STEMCELL Technologies, Cat# 04434) in triplicate. The vial was vigorously vortexed to thoroughly mix cells with media and then seeded in a 35 mm dish. The dishes were incubated for about 7–14 days at 37 °C and 5% $CO_2$. Colony-forming units (CFUs) were counted according to the manufacturer's guidelines. Cells from each plate were washed off and related 1000 cells per plate onto fresh methylcellulose media every 7 days in three rounds (Mentz et al, 2022).

## Isolation of CD34+ cells in cord blood and leukemia patient

The samples were mixed 1:1 with cold phosphate-buffered saline (PBS) buffer first, and then the CD34+ cells were isolated and purified using the Dynabeads™ CD34 Positive Isolation Kit (Invitrogen, USA). The purified CD34+ cells were cultured in StemSpan SFEM medium (StemCell Technologies) with 20% FBS,100 ng/mL SCF, 100 ng/mL FLT3, 100 ng/mL TPO, 20 ng/mL IL-3, and 20 ng/mL IL-6 to execute the following experiments (Weng et al, 2018).

## Subcellular fractionation

The subcellular fractionation was performed as described with some modifications (Gagnon et al, 2014; Han et al, 2022b). Briefly, 10 million cells were used for each sample. The cells were pretreated with or without EPZ5676 before collection and washed with ice-cold 1× PBS. Then the cells were lysed with HLB buffer to isolate the cytoplasmic fraction (supernatant) and the nuclear pellet. The nuclei pellet was further fractionated into nucleoplasmic

fraction (supernatant) and chromatin pellet by using MWS buffer. All the buffers above were supplemented with Halt™ Pretease inhibitor Cocktail (Thermo Fisher Scientific) and RNasin Ribonucleases Inhibitor (Promega). The RNA precipitation solution (RPS) buffer was added to the cytoplasmic fraction and nucleoplasmic fraction immediately, and each fraction was stored at −20 °C for at least 1 h. The two fractions were centrifuged, and the resulting pellets were washed with 75% ethanol. All the pellets from the chromatin fraction, nucleoplasmic fraction, and cytoplasmic fraction were added with 1 ml TRIzol reagent (Invitrogen) to extract RNA and protein. The protein pellets of each fraction were then dissolved in equal volumes of 1× loading buffer and used for western blot analyses.

## ChIP-qPCR

ChIP analyses were performed on chromatin extracts from MOLM-13 cells using a Magna ChIP G-Chromatin immunoprecipitation Kit (17-611, Merck Millipore) with primary antibody according to the manufacturer's standard protocol. In this assay, the fold enrichment of RBM15B and METTL3 was quantified via qRT-PCR and calculated relative to the input chromatin.

## In vitro pull-down assays

Plasmids expressing the recombinant proteins were transformed into Transetta (DE3) Chemically Competent cells (TransGen Biotech), and the protein expression was induced by IPTG (TransGen Biotech) (Fang et al, 2020; Sun et al, 2021). The pelleted bacteria were resuspended in PBS buffer supplemented with 1×complete ULTRA protease inhibitor (Roche) and then lysed by sonication. GST-tag and His-tag recombinant proteins expressed from pET-N-GST-Thrombin-C-His were purified using ProteinIso® GST Resin (TransGen Biotech) or Ni Sepharose® 6 Fast Flow (GE Healthcare). The purified GST- RBM15B-His was cleaved by thrombin (Sigma-Aldrich) to remove the GST tag.

For in vitro GST pull-down, an equal amount of GST-RBM15B-HIS recombinant proteins or GST/MBP control bound to glutathione sepharose/amylose resin was incubated with Recombinant Histone H3K79me2 (31221, Active Motif) or H3.3 (31295, Active Motif) for 1 h at 4 °C. After extensive washes, the proteins bound to the beads were suspended in 1×loading buffer (Fdbio science) and boiled for 10 min at 95 °C. The samples were analyzed by western blot.

## Polysome profiling

The polysome profiling was performed as described with some modifications (Han et al, 2022a; Sun et al, 2021; Sun et al, 2019). Cells were treated with cycloheximide (CHX; 200 μg/ml) for 15 min, and then $7 \times 10^7$ cells were lysed in lysis buffer (20 mM Tris-HCl, pH 7.4, 15 mM $MgCl_2$, 200 mM KCl, 1% Triton X-100) supplemented with 40U/ml RNasin (Promega), 1×complete ULTRA protease inhibitor (Roche), 100 μg/ml CHX, and 1 mM dithiothreitol (DTT). Cell lysates were centrifuged at 13,000 rpm at 4 °C for 10 min, and the supernatant was loaded onto 10% to 45% sucrose gradients followed by ultracentrifugation with a SW41 rotor (Beckman) at 36,000 rpm, 4 °C for 3 h. Absorbance at 260 nm was recorded using a BioComp Piston Gradient Fractionator

equipped with a Bio-Rad Econo UV Monitor. The corresponding isolated fractions were further used for RNA extraction with TRIzol reagent (Invitrogen). Extracted RNA samples were then used for qRT-PCR. Relative distribution of messenger RNA (mRNA) in each fraction was normalized by the total abundance of mRNA in all fractions, marked as 100%.

## Ribosome profiling (ribo-seq)

The control and RBM15B-depleted MOLM-13 cells were used for ribosome profiling. The procedure was carried out as previously described with a few modifications (Ingolia et al, 2012). Briefly, cells were treated with CHX at 100 μg/ml for 5 min before collection and lysis. The lysate was then treated with RNase I (NEB) and DNase I (NEB) for 45 min at room temperature. Nuclease digestion was stopped by adding SUPERase. Monosomes were separated by sucrose and centrifugation at 50,000 rpm at 4 °C for 1.5 h. The protected RNA fragments were isolated using the RNA Clean and Concentrator (Zymo Research). rRNA was further removed as previously described (Morlan et al, 2012). The purified RNA fragments were subsequently used for library construction with Next Multiple Small RNA Library Prep Set for Illumina (NEB). The constructed cDNA library was sequenced using Illumina HiseqTM X10 platform by Gene Denovo Biotechnology Co.

## Animal models

Five-week-old male NOD-SCID mice were maintained under pathogen-free conditions in the Laboratory Animal Center of Sun Yat-sen University, and subsequent procedures were performed in accordance with the institutional ethical guidelines for animal experiments. All animal studies were approved by the Institutional Animal Care and Use Committee of the Laboratory Animal Center of Sun Yat-sen University (approved No. SYSU-IACUC-2023-B0443).

The detailed procedures were performed as previously described (Wang et al, 2019). Briefly, a total of 100 μl of PBS or MOLM-13 cells ($3 \times 10^6$ cells) suspended in PBS for the corresponding group was injected through the tail vein for each mouse. Random groups of mice were sacrificed after 3 weeks. Human cell engraftment and infiltration (hCD45+ cell populations) in the bone marrow, peripheral blood, liver, and spleen were evaluated by flow cytometry or hematoxylin and eosin (H&E) staining performed as described. A part of the mice was euthanized for subsequent experiments, and the remaining mice were used for survival analysis. For animal treatment, mice were randomized and treated with 50 mg/kg EPZ5676, or vehicle (2% DMSO + 30% PEG 300 + 5% Tween 80 + 63% PBS) every other day for 10 days after 5 days of transplantation (Rau et al, 2016).

For patient-derived xenograft (PDX) experiments: primary cells were derived from *MLL*-r *(MLL-AF4)* and *MLL*-wt *(BCR-ABL1)* B-ALL patient samples, and infected with sh-NC or sh-RBM15B lentivirus before being injected into the tail vein of recipient NOD-SCID mice ($2 \times 10^6$ cells in 150 μL PBS per mouse). Random groups of mice were sacrificed after 3 weeks. The detailed procedures were performed as previously described (Heikamp et al, 2022; Yu et al, 2024). The engraftment and infiltration of PDX cell populations in the bone marrow, peripheral blood, liver, and spleen of the mice were evaluated by flow cytometry after about 8 weeks (*MLL*-r leukemia) or 12 weeks (*MLL*-wt leukemia). The rest of the mice

were used for survival analysis. Mouse bone marrow transplantation (BMT) of *MLL-AF9*-driven leukemia model was conducted as described previously (Liu et al, 2014; Yu et al, 2024). Leukemia cells isolated from primary *MLL-AF9* mice BM were transduced with murine *Rbm15b* and control shRNA lentivirus. Then, the cells were injected into the recipient mice (NOD-SCID) via tail vein injection.

## m⁶A-seq

For GLORI (glyoxal and nitrite-mediated deamination of unmethylated adenosines-sequencing): total RNA was extracted using TRIzol reagent (Invitrogen, 15596018), and mRNA was purified using Dynabeads Oligo(dT)$_{25}$ (Promega, Z5300). mRNA was fragmented by incubation at 94 °C for 4 min with fragment buffer (NEB, E6150S). Two hundred nanograms of fragmented RNA were ordered through RNA protection, deamination, and the deprotection process. Around 100 ng treated RNA was prepared for library construction and final sequencing. The detailed protocol was according to Yi lab's (Liu et al, 2023).

For MeRIP-seq: examining m⁶A modifications on individual genes according to the Magan MeRIP m⁶A kit (Millipore) manufacturer's instructions. The eluted m⁶A RNA was then removed rRNA by using rRNA depletion kit (Thermo Fisher Scientific) for subsequent qPCR analysis or library construction. The primers used in MeRIP RNA qPCR analysis are listed in Appendix Table 4.

## CUT&Tag-seq

RBM15B CUT&Tag were performed as previously described (Janssens et al, 2021; Kaya-Okur et al, 2019). In brief, $5 \times 10^4$ MOLM-13 cells were incubated with 10 μL pre-washed concanavalin A-coated magnetic beads at 25 °C for 10 min. Cells were resuspended in 80 μL buffer with 1 μg RBM15B antibody per sample, rotated at 4 °C overnight. After fragmenting the DNA and digesting the protein using Proteinase K, the DNA fragments were constructed into libraries and final sequencing.

## Molecular docking

In brief, we first identified the Domain Boundary. The finger domain of Menin (UniProt: O00255) comprises residues 400–610 and adopts an antiparallel *β-sheet* fold that forms a histone-binding cradle. Structural analysis of the Menin-H3K79me2 complex (PDB: 4GQ4) reveals that this domain directly engages the H3K79me2 peptide through a conserved basic pocket formed by residues R476, K480, and K483, which coordinate the methylated lysine via hydrogen bonding and hydrophobic interactions (Lin et al, 2023). Sequence alignment of human Menin and RBM15B was performed using Clustal Omega (https://www.ebi.ac.uk/Tools/msa/clustalo/), focusing on the defined domain boundaries. The structural validation and docking methodology are as follows: Protein structures were retrieved from the AlphaFold database (Abramson et al, 2024). Domain boundaries were mapped using AlphaFold-predicted structures and cross-referenced with Pfam functional domains. Prior to docking, structures were processed in PyMOL 2.5.3 to remove low-confidence regions. Global rigid docking was performed using ZDOCK 3.0.2 to predict H3K79me2 binding modes. Resultant models were visualized and analyzed in PyMOL 2.5.3.

## Data analysis

### MeRIP-seq analysis

To process the MeRIP-seq data, we first used Trim Galore (v0.6.10) to trim adapters from the raw sequence data. We then filtered out reads mapping to small noncoding RNAs (including rRNAs, miRNAs, snoRNAs, snRNAs, and MT_tRNAs) using Bowtie2 (v2.5.0) (Langmead and Salzberg, 2012) with default parameters. The remaining reads were aligned to the human genome (hg19) using Hisat2 (v2.2.1) (Kim et al, 2019) in very sensitive mode. After alignment, we converted the SAM files to BAM format using SAMtools (v1.3.1) (Danecek et al, 2021) and removed duplicate reads with Picard (v3.0.0) by running "MarkDuplicates --REMOVE_DUPLICATES true." Peak calling was performed on the BAM files using exomePeak2 (v1.6.1) (Meng et al, 2013), and the peak distribution was calculated using metaplotR (Olarerin-George and Jaffrey, 2017). Differential peak analysis was conducted using MACS2 (v2.2.9.1) (Zhang et al, 2008) based on the MeRIP alignments.

### GLORI analysis

GLORI analysis was basically conducted using the GLORI-tools pipeline (Shen et al, 2024). For the analysis, we used only R2 data. During preprocessing, we trimmed adapters twice using Cutadapt (v4.4) (Kechin et al, 2017) with the parameters "-q 20 --overlap 5 -e 0.1 --minimum-length 33 -a GGGGGGG -a CCCCCCC -a AAAAAAA -n 3" and "-q 20 --overlap 5 -e 0.1 --minimum-length 33 -a AGATCGGAAGAGCA-CACGTCTGAACTCCAGTCAC -a GATCGTCGGA -a CCTTGGCCGGGCCGTTTG -g TCCGACGATC -n 4." PCR duplicates were then removed using Seqkit (v2.5.1) (Shen et al, 2016) with the command "seqkit rmdup -s." In addition, the first 11 bp of each sequence were removed using Fastx Trimmer (v0.0.14). The remaining reads were processed using the GLORI-tools. After calling the m$^6$A sites, their distribution was analyzed with metaplotR.

### CUT&Tag-seq analysis

After trimming adapters and filtering low-quality reads using Trim Galore (v0.6.10), we mapped the processed sequencing reads to the human genome (hg19) using Bowtie2 (v2.5.0) with default parameters. Duplicate reads were removed with Picard MarkDuplicates (v3.0.0). Peak calling was performed on the deduplicated reads using MACS2 (v2.2.9.1) with the parameters "-g hs -q 0.01 --nomodel -B --keep-dup all -f BAMPE", followed by signal distribution analysis using Deeptools (v3.5.4) (Ramírez et al, 2016) computeMatrix and plotProfile function. Differential peak calling was then conducted with MACS2 bdgdiff. Combined the public CHIP-seq data (Wang et al, 2020b; Data ref: Wang et al, 2020b), the Spearman correlation between H3K79me2 and RBM15B was analyzed using Deeptools multiBigwigSummary and R (v4.1.1). The Venn diagram was generated using the R package ggvenn, and overlap calculations were performed with Bedtools intersectBed (v2.31.0) (Quinlan, 2014; Quinlan and Hall, 2010).

### RNA-seq analysis

For the RNA-seq data, we trimmed adapters using Trim Galore (v0.6.10) and filtered out reads mapping to small noncoding RNAs with Bowtie2 (v2.5.0). The filtered reads were then aligned to the hg19 genome using Hisat2 (v2.2.1). Gene expression quantification was performed using StringTie (v2.1.7) (Kovaka et al, 2019; Pertea et al, 2016), and differential expression analysis was conducted using DESeq2 (v1.38.3) (Love et al, 2014) in R.

### Ribo-seq analysis

For the ribo-seq data, only R1 reads were used for further analysis. Adapters were trimmed using Trim Galore (v0.6.10), and reads mapping to small noncoding RNAs were filtered out with Bowtie2 (v2.5.0). The remaining reads were aligned to the hg19 genome using Hisat2 (v2.2.1), and the alignments were sorted and converted to BAM format using SAMtools (v1.3.1). Gene-level read counts were quantified with StringTie (v2.1.7), and differential gene expression was analyzed using DESeq2 (v1.38.3) in R. Translation efficiency was assessed using Riborex (v2.4.0) (Li et al, 2017a), incorporating RNA-seq data.

### Other statistical analysis

We did not use any criteria to determine the sample size. As much data as possible was collected, depending on the nature of the experiments or in order to have a statistical analysis. Each experiment was performed with at least three biological replicates, as indicated in the figure legends. No blinding method was applied. For clinical data, the Mann–Whitney test was used to determine the significance of differentially expressed mRNA levels between the two groups; the non-parametric Kruskal–Wallis and LSD-t test analyses were adopted to compare the statistical differences among three groups. Leukemia-free survival was calculated from the date of complete remission (CR) until either relapse or death in remission, and analyzed using the Kaplan–Meier method with a log-rank test. For in vitro assays, the Shapiro–Wilk test was employed to assess the normality of the data. Values are the mean ± SEM of $n = 3$ independent experiments. Two-way ANOVA, one-way ANOVA, and unpaired two-tailed Student's $t$ test were used to determine the significance of differences. In this study, $P < 0.05$ was considered statistically significant.

## Data availability

The source data of this paper are collected in the following database record: NCBI GEO: GSE237261 and PRJNA1314999.

The source data of this paper are collected in the following database record: biostudies:S-SCDT-10_1038-S44318-026-00707-1.

## Peer review information

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

## Acknowledgements

We thank Prof. Yangqiu Li (Institute of Hematology, Medical College, Jinan University, China), Tao Cheng, and Hui Cheng (State Key Laboratory of Experimental Hematology, National Clinical Research Center for Blood Diseases, Chinese Academy of Medical Sciences and Peking Union Medical College, China) for providing the protocols for the *MLL-AF9*-driven mice model. We also thank Prof. Jingxuan Pan (State Key Laboratory of Ophthalmology, Zhongshan Ophthalmic Center, Sun Yat-sen University, China) for providing the protocols for primary cell culture and PDX model, and Cheng-Qi Yi (Peking-Tsinghua Center for Life Sciences, Peking University, China) for providing the help for GLORI analysis. This research was supported by the National Key R&D Program of China (No. 2022YFA1303302), Basic and Applied Basic Research Foundation of Guangdong Province (No. 2025B0303000012), National Natural Science Foundation of China (Nos. 32170570, 32370594, 32270598, 32400439), and Basic and Applied Basic Research Foundation of Guangdong Province (No. 2022A1515140018).

## Author contributions

**Tian-Qi Chen**: Data curation; Formal analysis; Writing—original draft. **Yu-Meng Sun**: Data curation; Formal analysis; Validation; Visualization; Methodology. **Shun-Xin Zhu**: Resources; Software; Formal analysis; Methodology; Writing—original draft. **Xiao-Tong Chen**: Data curation; Visualization; Methodology. **Ke-Jia Pu**: Data curation; Formal analysis. **Heng-Jing Huang**: Data curation; Visualization. **Qi Pan**: Data curation; Validation. **Jun-Yi Lian**: Data curation; Validation. **Wei Huang**: Conceptualization; Visualization. **Ke Fang**: Conceptualization; Visualization. **Xue-Qun Luo**: Data curation; Visualization. **Li-Bin Huang**: Formal analysis; Validation; Visualization; Writing—review and editing. **Yue-Qin Chen**: Conceptualization; Supervision; Investigation; Project administration;

Writing—review and editing. **Wen-Tao Wang**: Conceptualization; Supervision; Funding acquisition; Investigation; Methodology; Writing—original draft; Project administration; Writing—review and editing.

Source data underlying figure panels in this paper may have individual authorship assigned. Where available, figure panel/source data authorship is listed in the following database record: biostudies:S-SCDT-10_1038-S44318-026-00707-1.

## Disclosure and competing interests statement

The authors declare no competing interests.

