## [Peer Review File · The EMBO Journal]

RBM15B recognizes H3K79me2 to guide selective m6A-modification of mRNA and enhance oncoprotein translation in MLL-r leukemia

Tian-Qi Chen, Yu-Meng Sun, Shun-Xin Zhu, Xiao-Tong Chen, Ke-Jia Pu, Heng-Jing Huang, Qi Pan, Jun-Yi Lian, Wei Huang, Ke Fang, Xue-Qun Luo, Li-Bin Huang, Michelle Yue-Qin Chen, and Wen-Tao Wang

Corresponding authors: Michelle Yue-Qin Chen (lsscyq@mail.sysu.edu.cn) , Wen-Tao Wang (wangwt8@mail.sysu.edu.cn), Li-Bin Huang (huanglb3@mail.sysu.edu.cn)

Review Timeline:

Submission Date:	14th Apr 25
Editorial Decision:	23rd May 25
Revision Received:	19th Aug 25
Editorial Decision:	2nd Dec 25
Revision Received:	14th Dec 25
Accepted:	9th Jan 26

Editor: Daniel Klimmeck

Transaction Report:

Dear Dr Chen,

Thank you again for the submission of your manuscript (EMBOJ-2025-121070) to The EMBO Journal. Please accept my sincere apologies for getting back to you with this unusual delay, which is due to protracted referee input at this time as well as detailed discussion on your work in the editorial team. As mentioned earlier, your study was assessed by three reviewers with expertise in epitranscriptomics, protein translation and cancer biology, whose comments are enclosed below.

As you will see from the experts' reports, the referees acknowledge the analysis and potential interest and value of your findings. However, they also express important issues regarding the completeness of your study and rigor in supporting your claims, which need to be addressed thoroughly to make them supportive of publication in the EMBO Journal. Further, the reviewers raise a number of issues related to the presentation of the findings, additional controls and improved methods annotation required, statistics applied and overall discussion of related literature, that would need to be conclusively addressed to achieve the level of robustness and clarity needed for The EMBO Journal.

Given the overall interest stated and broader angle of your findings, we are able to invite you to revise your manuscript experimentally to address the referees' comments. I need to stress though that we do require strong support from the referees on a revised version of the study in order to move on to publication of the work.

Please feel free to contact me if you have any questions or need further input on the referee comments.

When submitting your revised manuscript, please carefully review the instructions below.

Please feel free to approach me any time should you have additional questions related to this.

Thank you for the opportunity to consider your work for publication.

I look forward to your revision.

Best regards,

Daniel Klimmeck

Daniel Klimmeck, PhD
Senior Editor
The EMBO Journal

Instruction for the preparation of your revised manuscript:

- 1) a .docx formatted version of the manuscript text (including legends for main figures, EV figures and tables). Please make sure that the changes are highlighted to be clearly visible.
- 2) individual production quality figure files as .eps, .tif, .jpg (one file per figure).
- 3) a .docx formatted letter INCLUDING the reviewers' reports and your detailed point-by-point response to their comments. As part of the EMBO Press transparent editorial process, the point-by-point response is part of the Review Process File (RPF), which will be published alongside your paper.
- 4) a complete author checklist, which you can download from our author guidelines ([https://wol-prod-cdn.literatumonline.com/pb-assets/embo-site/Author Checklist%20-%20EMBO%20J-1561436015657.xlsx](https://wol-prod-cdn.literatumonline.com/pb-assets/embo-site/Author%20Checklist%20-%20EMBO%20J-1561436015657.xlsx)). Please insert information in the checklist that is also reflected in the manuscript. The completed author checklist will also be part of the RPF.
- 5) Please note that all corresponding authors are required to supply an ORCID ID for their name upon submission of a revised

manuscript.

6) It is mandatory to include a 'Data Availability' section after the Materials and Methods. Before submitting your revision, primary datasets produced in this study need to be deposited in an appropriate public database, and the accession numbers and database listed under 'Data Availability'. Please remember to provide a reviewer password if the datasets are not yet public (see <https://www.embopress.org/page/journal/14602075/authorguide#datadeposition>).

7) Our journal encourages inclusion of *data citations in the reference list* to directly cite datasets that were re-used and obtained from public databases. Data citations in the article text are distinct from normal bibliographical citations and should directly link to the database records from which the data can be accessed. In the main text, data citations are formatted as follows: "Data ref: Smith et al, 2001" or "Data ref: NCBI Sequence Read Archive PRJNA342805, 2017". In the Reference list, data citations must be labeled with "[DATASET]". A data reference must provide the database name, accession number/identifiers and a resolvable link to the landing page from which the data can be accessed at the end of the reference. Further instructions are available at .

8) At EMBO Press we ask authors to provide source data for the main and EV figures. Our source data coordinator will contact you to discuss which figure panels we would need source data for and will also provide you with helpful tips on how to upload and organize the files.

Numerical data can be provided as individual .xls or .csv files (including a tab describing the data). For 'blots' or microscopy, uncropped images should be submitted (using a zip archive or a single pdf per main figure if multiple images need to be supplied for one panel). Additional information on source data and instruction on how to label the files are available at .

9) We replaced Supplementary Information with Expanded View (EV) Figures and Tables that are collapsible/expandable online (see examples in <https://www.embopress.org/doi/10.15252/embj.201695874>). A maximum of 5 EV Figures can be typeset. EV Figures should be cited as 'Figure EV1, Figure EV2' etc. in the text and their respective legends should be included in the main text after the legends of regular figures.

11) For data quantification: please specify the name of the statistical test used to generate error bars and P values, the number (n) of independent experiments (specify technical or biological replicates) underlying each data point and the test used to calculate p-values in each figure legend. The figure legends should contain a basic description of n, P and the test applied. Graphs must include a description of the bars and the error bars (s.d., s.e.m.).

We realize that it is difficult to revise to a specific deadline. In the interest of protecting the conceptual advance provided by the work, we recommend a revision within 3 months (21st Aug 2025). Please discuss the revision progress ahead of this time with the editor if you require more time to complete the revisions.

Referee #1:

The study presents a comprehensive mechanistic and translational study on the interplay between histone methylation and RNA methylation in MLL-rearranged leukemia (MLL-r).

The authors have performed a tour de force to identify the mechanism behind the connection between m6A and K79me, implicating RBM15 and the methyltransferase machinery, as well as relevant domains and residues on RBM15 that recruit METTL3.

Importantly, they show that one can target MLL-r leukemia cases via combination of RBM15 silencing and epigenetic inhibitors.

I have only a few questions that might help clarify aspects of the study:

-How is the RBM15 locus regulated at the transcriptional level? what do we know? Can it be autoregulated at the posttranscriptional level? Can the authors check datasets for RBM15 protein level changes?

-Do other epigenetic modifications change upon shRBM15 expression?

-What is the status of m6A in TWINS areas and RBM15 levels in hematopoietic progenitor cells?

-Are the MLLr leukemia cells more sensitive to FTO or METTL3 inhibitors compared to other leukemia types? I recommend testing FTO inhibitors and STM compounds. Does combination of STM and EPZ compounds act synergistically in inhibiting MLLr leukemias? Is YTHDC1 playing a role in this case, as it has been previously shown in the case of RBM15-mediated XIST methylation (PMID: 27602518).

-TWINS region specificity: Are there epitranscriptomics marks, other than m6A, that might decorate the TWINS loci?

-In the past, m6A levels decorating splicing factor transcripts have been shown to play critical roles in their translation (PMC10150290, PMID: 36944332). Have the authors noticed any changes in splicing factor transcript abundance and translation upon RBM15 silencing?

Referee #2:

In this study, Chen et al. identify RBM15B as a key regulator of N6-methyladenosine (m6A) modifications, specifically in the 5'untranslated regions (UTRs) and around the start codons of mRNAs. A significant finding is the revelation that this process is guided by H3K79me2, an epigenetic modification crucial in mixed lineage leukemia, with the H47 residue of RBM15B potentially mediating this recognition.

The authors propose that this H3K79me2-RBM15B axis-orchestrated m6A modification enhances the translation efficiency of oncogenic transcripts. This, in turn, is shown to be essential for the self-renewal of leukemic stem cells and the maintenance of leukemia. Crucially, blocking this axis was found to inhibit leukemia cell survival, promote differentiation, and impair hematological malignancies.

Overall, the paper presents a novel model for site-selective m6A deposition and highlights RBM15B as a promising therapeutic target for hematological malignancies. The findings are significant, offering both mechanistic insights into epigenetic regulation in tumorigenesis and potential avenues for therapeutic intervention. The study is well designed, and I think it is suitable for publication in the EMBO Journal if the authors can address to the points raised below.

Major Concerns:

1. While the authors emphasize the critical role of the H3K79me2-RBM15B axis in the translational control of specific oncogenes, the presented data do not sufficiently support this claim. Further experimental evidence is needed to solidify the direct link between this axis and the selective enhancement of oncogene translation.

a) While m6A is known to impact translation, the specific mechanism by which the H3K79me2-RBM15B axis selectively facilitates translation of oncogenic transcripts via m6A at start codons needs further elucidation and experimental backing. For example, if you mutate m6A sites, do you observe a decrease in TE?

b) How many of the transcripts identified in the ribosome profiling contain TWINS? Do you see a correlation between the position of the m6A and TE?

c) In addition to translationally downregulated transcripts, there are several upregulated mRNAs. What are those mRNAs? Do they contain m6A in their 5'UTR?

2. The authors consistently use a two-tailed Student's t-test to determine statistical significance across all figures. While appropriate for direct comparisons between two groups, this approach may not be suitable for all experimental designs presented. I recommend that the authors re-evaluate their statistical analyses and consider applying ANOVA where applicable, followed by appropriate post-hoc tests if necessary

Minor Concerns:

1. To ensure clarity and enable proper interpretation of the results, it is imperative that the authors clearly indicate which cell lines are MLL-wt and which are MLL-r in all relevant figures. Currently, the distinction between these crucial cell line types is not consistently clear, making it difficult to understand the context and implications of the presented data. Please amend all figure legends and, where appropriate, the figure panels themselves, to explicitly label the cell line status.

2. In Figure 1i, a notable observation is the clear increase in the m6A signal specifically located within the 3'UTRs. While the main focus of the manuscript is on m6A in 5'UTRs and around start codons, this 3'UTR m6A enrichment warrants further attention and discussion. The authors should address this finding. Specifically, do they observe any correlation between this alteration in 3'UTR m6A and changes in translational efficiency (TE), either an upregulation or downregulation, in those particular mRNAs? Investigating this could reveal additional layers of m6A-mediated gene regulation pertinent to the study's scope or at least provide a valuable clarification.

3. In Figure 2c, the authors state that there is a significant decrease in m⁶A in the TWINS region of target gene transcripts upon H3K79me2 suppression. However, this claim is not clearly supported by the figure as presented, and the visual evidence alone is insufficient. To substantiate this assertion, the authors must provide clear statistical analysis for the observed changes.

4. The IPs in the Supp Fig 5b has a very high background in the IgG control and make it difficult to interpret. Please provide a better IP.

5. Indicate the additional band in the Supp Fig 5f. Unspecific? Light chain?

6. To enhance reader comprehension and provide better context for the study's findings, it would be beneficial to include a small paragraph in the Introduction outlining what is currently known about RBM15. This would help to clearly establish the foundation upon which the current research builds and highlights the significance of identifying RBM15B's novel role.

7. Please provide an alignment of Menin finger domain and K79HMBD. It is unclear how you identified the residues.

8. The labels are missing for the panel 4a.

9. For the polysome profiling, it is common to show two conditions (Control vs KD) on the same graph to compare overall protein synthesis. Please merge the two panels in Fig 5f as well as in Fig 5g.

Point-by-point response to reviewers' comments and questions

We appreciate all the comments and criticisms.

Editor's comments:

Thank you again for the submission of your manuscript (EMBOJ-2025-121070) to The EMBO Journal. As you will see from the experts' reports, the referees acknowledge the analysis and potential interest and value of your findings. However, they also express important issues regarding the completeness of your study and rigor in supporting your claims, which need to be addressed thoroughly to make them supportive of publication in the EMBO Journal. Further, the reviewers raise a number of issues related to the presentation of the findings, additional controls and improved methods annotation required, statistics applied and overall discussion of related literature, that would need to be conclusively addressed to achieve the level of robustness and clarity needed for The EMBO Journal.

Given the overall interest stated and broader angle of your findings, we are able to invite you to revise your manuscript experimentally to address the referees' comments. I need to stress though that we do require strong support from the referees on a revised version of the study in order to move on to publication of the work.

Reply: Thank you very much for all the comments. In response to your suggestions, we have performed a number of additional experiments to address the reviewers' concerns and questions, including (1) applying the ChIP assays, western blots, and iBET151 treatment to demonstrate that MLL-FP/BRD4 complex could recognize H3K27ac and bind to RBM15B locus to activate RBM15B expression in *MLL-r* leukemia; (2) Detecting the status of m⁶A in TWINS areas and RBM15B levels in normal or *MLL-r* leukemia CD34+ cells; (3) Investigating the roles of FTO or METTL3 inhibitors in *MLL-r* leukemia and verifying whether combination of STM2457 and EPZ5676 compounds could restrain *MLL-r* leukemia; (4) Investigating whether knocking down RBM15B can affect other epigenetic modifications; (5) Constructing two *GFP* reporter systems including *TWINS^{m6Awt}-GFP* and *TWINS^{m6Amut}-GFP* to investigate the m⁶A role of TWINS in translation, which was regulated by dysregulation of RBM15B or EPZ5676 treatment; (6) Performing a serial of re-analysis suggested by the two reviewers. We have also addressed other issues raised by the two reviewers. Please find below our point-by-point responses. In addition, we have also add a reviewer token for the deposited GSE237261 dataset: qhupioukdbaphaz.

The input from you and the colleagues has helped greatly improve our manuscript. Thank you.

Referee #1:

The study presents a comprehensive mechanistic and translational study on the interplay between histone methylation and RNA methylation in MLL-rearranged leukemia (MLL-r).

The authors have performed a tour de force to identify the mechanism behind the connection between m6A and K79me, implicating RBM15 and the methyltransferase machinery, as well as relevant domains and residues on RBM15 that recruit METTL3. Importantly, they show that one can target MLL-r leukemia cases via combination of RBM15 silencing and epigenetic inhibitors.

I have only a few questions that might help clarify aspects of the study:

Comment 1 -How is the RBM15 locus regulated at the transcriptional level? what do we know? Can it be autoregulated at the posttranscriptional level? Can the authors check datasets for RBM15 protein level changes?

Reply: We appreciate your insightful comment. Following your suggestion, we have performed additional analysis and experiments to investigate the regulatory mechanism underlying higher expression of RBM15B in *MLL*-r leukemia. Specific as follows:

(1) Previous studies have revealed that MLL fusion protein could mediate transcription regulation mainly dependent on the formation of MLL-FP complex, including MLL-FP/DOT1L complex-H3K79me axis or MLL-FP/BRD4 complex-H3K27ac axis. In the current study, we found that downregulation of H3K79me by EPZ5676 did not affect RBM15B expression (shown in Appendix Figure S2G), indicating that RBM15B expression might not be regulated by MLL-FP/DOT1L complex-H3K79me axis. We asked if MLL-FP/BRD4 complex might regulate RBM15B expression. Thus, we reanalyzed the MLL-FP Cut&Tag, BRD4 and H3K27ac ChIP datasets (Erb et al, 2017; Data ref: Erb et al, 2017; Gilan et al, 2016; Data ref: Gilan et al, 2016; Janssens et al, 2021; Data ref: Janssens et al, 2021), and found that high enrichment of H3K27ac, MLL-FP and BRD4 indeed existing in *RBM15B* locus (**New Appendix Figure S1C**).

(2) To validate the analysis, we treated the *MLL*-r leukemia cells by iBET151. The qRT-PCR and western blot data showed that iBET can inhibit the RBM15B expression (**New Appendix Figure S1D**). Moreover, we found that BRD4 inhibitor

(iBET151) can significantly suppress the enrichment of BRD4 on RBM15B locus (**New Appendix Figure S1C**). Knockdown of MLL-AF9 can reduce *RBM15B* expression (**New Appendix Figure S1E**). We have also performed additional ChIP assays with MLL and BRD4 antibodies, and found that MLL and BRD4 enriched on RBM15B locus and these enrichments can be destroyed by iBET151, further supporting that MLL-FP/BRD4 complex recognizes H3K27ac and binds to RBM15B locus to activate RBM15B expression (**New Appendix Figure S1F**). These new data suggest that MLL-FP/BRD4 complex located at the *RBM15B* locus and regulate its expression.

(3) To demonstrate whether mRNA stability of RBM15B also affects the RBM15B protein level, we have compared the stability of RBM15B mRNAs in *MLL*-wt (NB4) and *MLL*-r leukemia cells (MOLM-13), no significant difference on RBM15B mRNA stability between these two-type leukemia was found (**Review Figure 1A**), suggesting that the RBM15B may not be autoregulated at the posttranscriptional level. As suggested, we have searched the published datasets and obtained a proteomic dataset of AML (Kramer et al, 2022) were found, which only has 4 *MLL*-r cases. We analyzed the dataset, RBM15B protein levels showed slightly higher in *MLL*-r AML samples (**Review Figure 1B**), however, due to the small number of *MLL*-r AML samples and big error bar in the *MLL*-wt samples, the difference was not significant. We have applied 8 *MLL*-r samples together with *MLL*-wt patient and healthy samples for detection. As shown in Figure 1C, higher RBM15B protein levels were detected in *MLL*-r leukemia samples than those in *MLL*-wt and healthy samples (Figure 1C).

These additional analysis and experimental data together suggest that MLL-FP/BRD4 complex recognizes H3K27ac and binds to *RBM15B* locus to activate *RBM15B* expression in *MLL*-r leukemia. This possible regulation mechanism may explain the higher expression of RBM15B in *MLL*-r leukemia. We appreciate the suggestion to make the statement clear. These new data have been added into the revised manuscript (**pages 6, lines 146-165**).

Figure legend: **New Appendix Figure S1C** Genome browser views of ChIP-seq profile of H3K27ac in MV4-11(GSE82116), Cut&Tag profile of MLL-AF9 in MPAL1 and MLL-AF4 in RS4;11(GSE159608), ChIP-seq profile of BRD4 (GSE71780) at *RBM15B* gene locus in iBET151 treated compared to DMSO control. **New Appendix Figure S1D** Relative RNA expression and protein level of *RBM15B* treated with iBET151 or DMSO for 48 hours. **New Appendix Figure S1E** Western blots showing the MLL-AF9, *RBM15B* protein levels under knockdown of MLL-AF9. GAPDH regarded as internal control. **New Appendix Figure S1F** ChIP-qPCR of BRD4 and MLL binding at *RBM15B* gene loci treated with iBET151. **Review Figure 1A** RNA stability of *RBM15B* transcripts detected by qRT-PCR in MOLM-13 and NB4 cells. 10 μ g/ml Actinomycin D was used in the assays. **Review Figure 1B** Analysis of *RBM15B* protein level in the published proteomic dataset of AML (Kramer et al, 2022).

Comment 2. Do other epigenetic modifications change upon shRBM15 expression?

Reply: According to your comment, we have performed additional experiments to investigate whether other epigenetic modifications change upon sh-RBM15B expression:

(1) First, we checked a set of popular epigenetic modifications including H3K4me2/3, H3K36me2/3, H3K27me2/3, and H3K27ac in *MLL-r* leukemia cells electrotransfected with si-NC or si-RBM15B. The result showed that knockdown of RBM15B did not show significantly effects on these epigenetic modifications detected (**Review Figure 2A**).

(2) We have also investigated whether shRBM15B can affect other RNA epigenetic modification such as m5C via performing m5C meRIP assay under sh-RBM15B or sh-NC. In the m5C meRIP assay, the SHMT2 mRNA acted as the positive of m5C modification (Li et al, 2025). The results showed that m5C modifications are enriched in the TWINS regions of HOXA9, MYC, and BCL2; however, these modifications were not affected by shRBM15B (**Review Figure 2B**). These data suggest that RBM15B might not directly regulate many well-characterized epigenetic modifications but m⁶A modification. Thank you.

Figure legend: **Review Figure 2A** Western blot showing the H3K4me2/3, H3K36me2/3, H3K27me2/3, H3K27ac protein level in the RBM15B knockdown in MV4-11 cells. **Review Figure 2B** m5C meRIP-qPCR showing the positive control of SHMT2 transcripts. m5C meRIP-qPCR for TWINS region of BCL2, MYC and HOXA9 in the RBM15B knockdown in MOLM-13 cells. IgG as the negative control.

Comment 3. What is the status of m⁶A in TWINS areas and RBM15 levels in hematopoietic progenitor cells?

Reply: To address the comment on the status of m⁶A in TWINS areas and RBM15B levels in hematopoietic stem/progenitor cells, we have performed following two additional experiments.

(1) First, we isolated the hematopoietic stem/progenitor cells by CD34+ beads from umbilical cord blood of a recruited health infant and an *MLL-r* leukemia patient (Patient #8) (the method as showed in Fig. 4H). We found that RBM15B protein level was higher in *MLL-r* CD34+ cells than that in normal ones (**New Appendix Figure S16A**).

(2) Next, we performed the m⁶A meRIP in the RNAs from the normal CD34+ cells and *MLL-r* CD34+ cells. The result showed that m⁶A modifications are enriched in the TWINS regions of HOXA9, MYC, and BCL2 in both CD34+ samples. We also found that *MLL-r* CD34+ cells have higher m⁶A modifications in the TWINS regions of HOXA9 and BCL2 than those in normal ones (**New Appendix Figure S16B**). There was no change of m⁶A modifications in the TWINS regions of MYC between *MLL-r* and normal CD34+ cells. However, downregulation of RBM15B did not change the expression of HOXA9, MYC and BCL2, and suggesting that it may not play a role in normal hematopoietic progenitor cells.

These additional data suggest that m⁶A modifications within TWINS regions are important for *MLL-r* hematopoietic stem/progenitor cell function, which guided by H3K79me2–RBM15B–m⁶A axis. Furthermore, elevated RBM15B levels promote greater m⁶A enrichment in these regions, accelerating *MLL-r* leukemia progression. We appreciate the comment, and the new data have been added into the revised manuscript (**pages 18, lines 504-506**).

Figure legend: **New Appendix Figure S16A** The protein level of RBM15B in healthy CD34+ and *MLL-r* CD34+ cells. **New Appendix Figure S16B** m⁶A meRIP-qPCR for TWINS region of HOXA9, BCL2 and MYC compared in healthy CD34+ and *MLL-r* CD34+ cells.

Comment 4. Are the MLLr leukemia cells more sensitive to FTO or METTL3 inhibitors compared to other leukemia types? I recommend testing FTO inhibitors and STM compounds. Does combination of STM and EPZ compounds act synergistically in inhibiting MLLr leukemias? Is YTHDC1 playing a role in this case, as it has been previously shown in the case of RBM15-mediated XIST methylation (PMID: 27602518).

Reply: According to your comments, we have performed three additional experiments to answer the questions. Specific experiments and results are as follows:

(1) We treated the *MLL*-r leukemia cells (MV4-11, MOLM-13, and THP1) and *MLL*-wt leukemia cells (HL-60, NB4, and U937) by STM2457 (a METTL3 inhibitor) or FB23-2 (a FTO inhibitor). IC₅₀ assays showed that, compared to *MLL*-wt leukemia cells, *MLL*-r leukemia cells were sensitive to STM2457 (a METTL3 inhibitor) but not to FB23-2 (a FTO inhibitor) (**New Appendix Figure S15A-B**), suggesting that m⁶A modifications mediated by METTL3 complex have important functions in *MLL*-r leukemia.

(2) We further treated the *MLL*-r leukemia cells by the combination of STM2457 and EPZ5676 compounds. The CCK-8 assays showed that combination of STM2457 and EPZ5676 compounds can significantly suppress the *MLL*-r leukemia cell viability, the effect is better than that of STM2457 or EPZ5676 treatment alone (**New Appendix Figure S15C-D**). These data suggested that inhibition of the H3K79me-RBM15B/METTL3 complex-m⁶A axis by combination of STM2457 and EPZ5676 compounds can dramatically restrain *MLL*-r leukemia. We appreciate the comment, and the new data have been added into the revised manuscript (**pages 17-18, lines 489-499**).

(3) In the XIST methylation case, YTHDC1 preferentially recognizes the RBM15/15B-mediated m⁶A residues on XIST to mediate gene silencing (Patil et al, 2016), indicating a special role of YTHDC1 in gene silencing. To investigate whether YTHDC1 also play a role in RBM15B-mediated m⁶A in *MLL*-r leukemia, we reanalyzed the YTHDC1 iCLIP-seq data (PMID: 27602518) and took MYC, HOXA9 and BCL2 as examples. We found that YTHDC1 locates at MYC gene body and mainly enriched on stop coding regions, but barely locates at HOXA9 and BCL2 gene bodies (**Review Figure 3A**), suggesting that YTHDC1 may play a role in MYC expression but not HOXA9 and BCL2. In addition, we implied YTHDC1 knockdown in *MLL*-r leukemia cells. The result showed that YTHDC1 knockdown can slightly affect HOXA9 and BCL2 expression, these *loci* have considerable m⁶A modification in TWINS region in *MLL*-r leukemia cells (**Review Figure 3B**). We also observed a

significant decrease of MYC expression under YTHDC1 knockdown (**Review Figure 3B**). MYC is oncogene that widely drives many types of cancers including leukemia (Cheng et al, 2021; Lourenco et al, 2021). Previous study found that YTHDC1 can regulate MYC expression by reading the m⁶A modification of stop coding regions (Cheng et al, 2021). These data may suggest that YTHDC1 played a role mainly via reading m⁶A modification of stop coding regions.

All these data together suggest that YTHDC1 might not play a direct role in the H3K79me-RBM15B-m⁶A axis in *MLL-r* leukemia. Hope that these additional data have addressed your questions and concerns. Thank you very much.

Figure legend: **New Appendix Figure S15A and B** STM2457 (A) and FB23-2 (B) dose-response curve for a panel of leukemia cell lines. IC₅₀ for each cell line is shown in brackets and calculated by GraphPad prism; **New Appendix Figure S15C and D** Proliferation of MOLM-13 (C) and MV4-11 (D) in four groups: DMSO, STM2457 (3 μM for MOLM-13, 5 μM for MV4-11), EPZ5676 (2 μM), and STM2457+EPZ5676. Data are means ± SEM of three biological replicates and were analyzed by two-way ANOVA (** *P* < 0.01, *** *P* < 0.001); **Review Figure 3A** Analysis of iCLIP-seq profile of YTHDC1 in MYC, HOXA9, and BCL2 transcripts; **Review Figure 3B** The protein level of YTHDC1, HOXA9, BCL2 and MYC in the YTHDC1 knockdown MV4-11 cells. GAPDH acted as the internal control.

Comment 5. TWINS region specificity: Are there epitranscriptomics marks, other than m6A, that might decorate the TWINS loci?

Reply: Thank you for your comment. Following the suggestion, we have performed additional experiments to investigate whether other RNA epigenetic modification, such as m5C, located at the TWINS loci (please also see our reply to your comment 2). We have performing m5C meRIP assay under sh-RBM15B or sh-NC. In the m5C meRIP assay, the SHMT2 mRNA acted as the positive of m5C modification (Li et al, 2025). The results showed that m5C modifications are also enriched in the TWINS regions of HOXA9, MYC, and BCL2; however, these modifications are not affected by shRBM15B (**Review Figure 2B**). These data suggest that RNA m5C modification also decorates the TWINS loci but this modification was not affected by RBM15B. Thank you.

Figure legend: Review Figure 2B m5C meRIP-qPCR showing the positive control of SHMT2 transcripts. m5C meRIP-qPCR for TWINS region of BCL2, MYC and HOXA9 in the RBM15B knockdown in MOLM-13 cells. IgG as the negative control.

Comment 6. In the past, m6A levels decorating splicing factor transcripts have been shown to play critical roles in their translation (PMC10150290, PMID: 36944332). Have the authors noticed any changes in splicing factor transcript abundance and translation upon RBM15 silencing?

Reply: We appreciate your insightful comment. As suggested, we searched for transcripts of splicing factors (304 genes listed) (Rogalska et al, 2024) in our Ribo-seq and RNA-seq data. 286 splicing genes were found in our study. Analysis revealed that transcript levels for most splicing factors (all except 1 genes) remained unchanged upon RBM15B silencing, suggesting that RBM15B does not affect the abundance of most splicing factor transcripts (**Review Figure 4**). In the translation analysis, however, we observed significant changes: 137 genes exhibited upregulated

translation ($\log_2 \text{FC} > 0.15$, $\text{FDR} < 0.05$), 44 genes showed downregulated translation ($\log_2 \text{FC} < -0.15$, $\text{FDR} < 0.05$), and 105 genes displayed no significant change in TE in the RBM15B-silenced samples (**Review Figure 4**). These data suggest that knocking down RBM15B may upregulate the translation of a set of splicing factor transcripts, but the mechanism requires further investigation. Thank you very much.

Figure legend: Review Figure 4 Scatterplot of the 286 splicing factor transcripts average fold change (FC) of ribosome and mRNA abundance in RBM15B-depleted MOLM-13 cells. (Ribo-seq, $\log_2 \text{FC} > 0.15$ and < -0.15 , $\text{FDR} < 0.05$; RNA-seq, $\log_2 \text{FC} > 0.5$ and < -0.5 , $\text{FDR} < 0.05$).

Referee #2:

In this study, Chen et al. identify RBM15B as a key regulator of N6-methyladenosine (m6A) modifications, specifically in the 5'untranslated regions (UTRs) and around the start codons of mRNAs. A significant finding is the revelation that this process is guided by H3K79me2, an epigenetic modification crucial in mixed lineage leukemia, with the H47 residue of RBM15B potentially mediating this recognition.

The authors propose that this H3K79me2-RBM15B axis-orchestrated m6A modification enhances the translation efficiency of oncogenic transcripts. This, in turn, is shown to be essential for the self-renewal of leukemic stem cells and the maintenance of leukemia. Crucially, blocking this axis was found to inhibit leukemia cell survival, promote differentiation, and impair hematological malignancies.

Overall, the paper presents a novel model for site-selective m6A deposition and highlights RBM15B as a promising therapeutic target for hematological malignancies. The findings are significant, offering both mechanistic insights into epigenetic regulation in tumorigenesis and potential avenues for therapeutic intervention. The study is well designed, and I think it is suitable for publication in the EMBO Journal if the authors can address to the points raised below.

Major Concerns:

1. While the authors emphasize the critical role of the H3K79me2-RBM15B axis in the translational control of specific oncogenes, the presented data do not sufficiently support this claim. Further experimental evidence is needed to solidify the direct link between this axis and the selective enhancement of oncogene translation.

Comment a) While m⁶A is known to impact translation, the specific mechanism by which the H3K79me2-RBM15B axis selectively facilitates translation of oncogenic transcripts via m⁶A at start codons needs further elucidation and experimental backing. For example, if you mutate m⁶A sites, do you observe a decrease in TE?

Reply: This is a very good comment indeed. To address the question and concern, we have performed three additional experiments:

(1) We constructed two *GFP* reporter systems: one is *TWINS*^{m⁶Awt}-*GFP* (DNA sequences containing the native promoter and TWINS region with m⁶A sites from *HOXA9* or *MYC* fused to the *GFP* sequence) and another one is *TWINS*^{m⁶Amut}-*GFP* (the native promoter and TWINS region with m⁶A mutation sites from *HOXA9* or *MYC* fused to the *GFP* sequence) (shown in **New Figure 5K**). The m⁶A sites were identified by GLORI in this study. These *GFP* reporter systems were integrated into DNA via lentivirus in MV4-11 or MOLM-13 cells. We found that *TWINS*^{m⁶Awt}-*GFP* of *HOXA9* or *MYC* constructs displayed significantly GFP⁺ ratio than control in the flow cytometry results (**New Appendix Figure S10A**), demonstrating the success of *GFP* reporter system constructed to indicate GFP translation in *TWINS*^{m⁶Awt}-*GFP* of *HOXA9* or *MYC* constructs.

(2) We next manipulated the RBM15B expression levels via transfecting CMV-RBM15B plasmid or si-RBM15B into *TWINS*^{m⁶Awt}-*GFP* or *TWINS*^{m⁶Amut}-*GFP* constructed cells, respectively to upregulate and/or downregulate RBM15B expression (**New Appendix Figure S10B and New Appendix Figure S10E**). The results confirmed that *TWINS*^{m⁶Awt}-*GFP* cells exhibited higher GFP translation levels than *TWINS*^{m⁶Amut}-*GFP* cells, supporting the role of m⁶A modifications within the RNA's TWINS region in enhancing translation (**New Figure 5L and New Appendix Figure S10C-D**). We also found that upregulation of RBM15B significantly enhanced GFP translation in *TWINS*^{m⁶Awt}-*GFP* cells, while downregulation of RBM15B reduced it. However, modulating RBM15B expression had limited influence on GFP translation in *TWINS*^{m⁶Amut}-*GFP* cells (**New Figure 5L-M and New Appendix Figure S10C-I**). These new data suggest that m⁶A within the TWINS region plays an important role in the translation of oncogenic transcripts and that RBM15B selectively facilitates this process.

(3) To investigate the role of DOT1L/H3K79me in the translation of oncogenic transcripts mediated by m⁶A within the TWINS region, we further treated the constructed *TWINS*^{m6Awt}-GFP or *TWINS*^{m6Amut}-GFP cells with the DOT1L inhibitor EPZ5676 (New Appendix Figure S11A). Previous studies have demonstrated that EPZ5676 decreases H3K79me2 modification at the loci of *MLL*-r target genes, and reduces their transcription levels and expression (Stein et al, 2018; Yi & Ge, 2022). Indeed, we found that EPZ5676 significantly reduced GFP translation in both *TWINS*^{m6Awt}-GFP and *TWINS*^{m6Amut}-GFP cells (New Appendix Figure S11B-E). Notably, EPZ5676 treatment dramatically reduced GFP translation by over 5-fold in *TWINS*^{m6Awt}-GFP of *HOXA9* cells and 2- to 5-fold in *TWINS*^{m6Awt}-GFP of *MYC* cells, compared to a reduction of 2-fold in *TWINS*^{m6Amut}-GFP of *HOXA9* cells and 1.5-fold in *TWINS*^{m6Amut}-GFP of *MYC* cells. These findings suggest that H3K79me2 not only regulates target gene transcription but also enhances the translation of oncogenic transcripts by facilitating RBM15B-mediated selective m⁶A site modification.

Together, these new data may further support the H3K79me2-RBM15B axis that selectively facilitates translation of oncogenic transcripts via m⁶A at start codons. We appreciate the comment, and the new data have been added into the revised manuscript (pages 15-16, lines 415-447).

Figure legend: New Figure 5K Schematic for system. New Appendix Figure S10A Representative flow cytometry images of GFP translation, demonstrating a successful GFP reporter system to indicate GFP translation in *TWINS*^{m6Awt}-GFP of *HOXA9* or *MYC* constructs.

New Appendix Figure S10B Western blots showed the upregulation RBM15B via transfecting CMV-RBM15B plasmid into *TWINS^{m6Awt}-GFP* or *TWINS^{m6Amut}-GFP* constructed MV4-11 (*MLL-AF4*) cells, respectively. (**New Appendix Figure S10C, D**) Representative flow cytometry images and statistical analysis of GFP translation in *TWINS^{m6Awt}-GFP* or *TWINS^{m6Amut}-GFP* (C for *HXOA9*, D for *MYC*) constructed MV4-11 (*MLL-AF4*) cells transfecting with or without CMV-RBM15B. Values are mean \pm SEM (n = 3), one-way ANOVA. ***P* < 0.01; ****P* < 0.001. **New Figure 5L** Statistical analysis of GFP translation in *HXOA9-TWINS^{m6Awt}-GFP* or *HXOA9-TWINS^{m6Amut}-GFP*. Values are mean \pm SEM (n = 3), one-way ANOVA. NS, not significant; ****P* < 0.001.

Figure legend: Appendix Figure S10E Western blots showed the downregulation of RBM15B via transfecting si-RBM15B into *TWINS^{m6Awt}-GFP* of *HXOA9* / *MYC* or *TWINS^{m6Amut}-GFP* of *HXOA9* / *MYC* constructed MV4-11 (*MLL-AF4*) or MOLM-13 (*MLL-AF9*) cells, respectively. **Appendix Figure S10G-I** Representative flow cytometry images and statistical analysis of GFP translation in *TWINS^{m6Awt}-GFP* of *HXOA9* / *MYC* or *TWINS^{m6Amut}-GFP* of *HXOA9* / *MYC* constructed MV4-11 (*MLL-AF4*) or MOLM-13 (*MLL-AF9*) cells transfecting with or without si-RBM15B. **New Figure 5M** Statistical analysis of GFP translation in *HXOA9-TWINS^{m6Awt}-GFP* or *HXOA9-TWINS^{m6Amut}-GFP*. Values are mean \pm SEM (n = 3), one-way ANOVA. NS, not significant; ***P* < 0.01; ****P* < 0.001.

Figure legend: Appendix Figure S11A Western blots showed the downregulation H3K79me2 by EPZ5676 treatment (1 μ M, 72 hours) in *TWINS*^{m6Awt}-GFP or *TWINS*^{m6Amut}-GFP constructed MV4-11 (*MLL-AF4*) or MOLM-13 (*MLL-AF9*) cells, respectively. (**Appendix Figure S11B-E**) Representative flow cytometry images and statistical analysis of GFP translation in *TWINS*^{m6Awt}-GFP of *HOXA9* / *MYC* or *TWINS*^{m6Amut}-GFP of *HOXA9* / *MYC* constructed MV4-11 (*MLL-AF4*) or MOLM-13 (*MLL-AF9*) cells treating with or without EPZ5676 (1 μ M, 72 hours). Values are mean \pm SEM (n = 3), one-way ANOVA. **P* < 0.05; ***P* < 0.01; ****P* < 0.001.

Comment b) How many of the transcripts identified in the ribosome profiling contain TWINS? Do you see a correlation between the position of the m6A and TE?

Reply: We appreciate your comment In the study, we defined this region "TWINS" (two wings around start codon), specifically referring to the regions around start codon within the transcripts of MLL-fusion protein target genes, and m⁶A modification in the regions is regulated by RBM15B. Therefore, we mainly considered those transcripts containing TWINS region and regulated by H3K79me2-RBM15B-m⁶A axis. According to your suggestion, we have researched the Ribo-seq dataset, RBM15B CUT&Tag dataset, H3K79me2 dataset, and

MeRIP-seq datasets. We found that, among the 1781 genes with downregulation of translation mediated by sh-RBM15B, 250 genes displayed the loss of H3K79me2 (ChIP-seq), RBM15B binding (CUT&Tag), and m⁶A marks (both of sh-RBM15B and EPZ5676) in both RBM15B knockdown and EPZ5676 treatment compared to the negative control samples, suggesting that they may be regulated by EPZ5676-RBM15B-m⁶A-axis in the TWINS and mediated translation.

Regarding to the correlation between the position of the m⁶A and TE, we did not find a significant correlation between the position of the m⁶A and TE in global genes. However, when only investigated those 250 genes, the downregulation of m⁶A in the TWINS suppress the TE of transcripts (57.2 %). Hope that our explanation is clear and accepted. Thank you very much.

c) In addition to translationally downregulated transcripts, there are several upregulated mRNAs. What are those mRNAs? Do they contain m⁶A in their 5'UTR?

Reply: We understand your concerns. Indeed, in addition to translationally downregulated transcripts, there are also upregulated mRNAs. Following your suggestion, we have analyzed these translationally upregulated genes. Gene Ontology (GO) showed that the top GO terms are “Nucleic acid metabolic process”, “RNA metabolic process” and “RNA biosynthetic process”, suggesting that these genes may contribute to RNA metabolic process. We also researched the m⁶A modification of their 5' UTRs in our GLORI datasets from the sh-NC and DMSO control groups. Only a small subset of genes had m⁶A in their 5' UTRs (4.1% in shNC_GLORI or 3.8% in DMSO_GLORI). This analysis may suggest that the translation of these upregulated mRNAs is not regulated by the H3K79me2-RBM15B-m⁶A axis in TWINS region, and they are not MLL-r targets. Thank you very much to make the data clearer.

Figure legend: Top GO terms analyzed within translationally upregulated mRNAs.

Comment 2. The authors consistently use a two-tailed Student's t-test to determine statistical significance across all figures. While appropriate for direct comparisons between two groups, this approach may not be suitable for all experimental designs presented. I recommend that the authors re-evaluate their statistical analyses and consider applying ANOVA where applicable, followed by appropriate post-hoc tests if necessary

Reply: We agree with your comment and have re-evaluated all the statistical analyses and applied ANOVA in Figure 1D, 2G, 4A, 4C-F, 5L-M, 6C, 6F, 7A,7D, New Appendix Figure S1D, S1I, S3I, S5C, S6A, S6C-D, S6F-H, S9E-F, S10D, S10F, S10H-I, S11B-E, S12D, S15C-D, S16D-E, and clearly indicate the method in Figure legends. Thank you very much.

Minor Concerns:

1.To ensure clarity and enable proper interpretation of the results, it is imperative that the authors clearly indicate which cell lines are MLL-wt and which are MLL-r in all relevant figures. Currently, the distinction between these crucial cell line types is not consistently clear, making it difficult to understand the context and implications of the presented data. Please amend all figure legends and, where appropriate, the figure panels themselves, to explicitly label the cell line status.

Reply: Thank you for pointing this out. We have described the cell lines *MLL*-wt and *MLL*-r in material method section as suggested. In addition, the cells have been labelled in the Figure at the suitable positions and described in the figure legend according to the comments, thanks a lot.

2. In Figure 1i, a notable observation is the clear increase in the m⁶A signal specifically located within the 3'UTRs. While the main focus of the manuscript is on m⁶A in 5'UTRs and around start codons, this 3'UTR m⁶A enrichment warrants further attention and discussion. The authors should address this finding. Specifically, do they observe any correlation between this alteration in 3'UTR m⁶A and changes in translational efficiency (TE), either an upregulation or downregulation, in those particular mRNAs? Investigating this could reveal additional layers of m⁶A-mediated gene regulation pertinent to the study's scope or at least provide a valuable clarification.

Reply: We appreciate your insightful comment. Indeed, we observed an increase of m⁶A signal density in the distal-end of 3' UTR. Following your comment, we have reanalyzed the meRIP-seq and Ribo-seq of the transcripts with elevated 3' UTR m⁶A to examine the changes in translational efficiency (TE). The result showed that 1046 (29572 in total genes) total transcripts and 40 (144 in total *MLL-r* targets) *MLL-r* targets were observed with increase of m⁶A modification in the distal-end of 3' UTRs (**Review Figure 6A and B**). In addition, we observed 187 (11 *MLL-r* targets) genes exhibited upregulated translation ($\log_2 \text{FC} > 0.15$, $\text{FDR} < 0.05$), 237 (12 *MLL-r* targets) genes showed downregulated translation ($\log_2 \text{FC} < -0.15$, $\text{FDR} < 0.05$), and 603 (17 *MLL-r* targets) genes displayed no significant change in TE in the RBM15B-silenced samples, (**Review Figure 6A and B**). These data suggest that RBM15B knockdown also elevates m⁶A signals in many distal-end of 3' UTRs and may affect the translation of multiple transcripts; however, this regulatory effect appears difference in different transcripts. Previous studies showed that 3' UTR m⁶A can promote or inhibit translation, and this effect is dependent on the modified specific location, cell environment, and the protein type (Corovic et al, 2025; Meyer, 2019). Our analysis also showed that the positions of m⁶A on different transcripts may have different roles on translation. We have discussed this in the Discussion section (**page 20, lines 569-575**). Thank you very much.

Figure legend: Scatterplot of the 1046 genes (**Review Figure 6A**) and 40 *MLL-r* targets genes

(Review Figure 6B) with average foldchange (FC) of ribosome and mRNA abundance in RBM15B-depleted MOLM-13 cells (ribo-seq, $\log_2 FC > 0.15$ and < -0.15 , FDR < 0.05 , RNA-seq, $\log_2 FC > 0.5$ and < -0.5 , FDR < 0.05).

3. In Figure 2c, the authors state that there is a significant decrease in m⁶A in the TWINS region of target gene transcripts upon H3K79me2 suppression. However, this claim is not clearly supported by the figure as presented, and the visual evidence alone is insufficient. To substantiate this assertion, the authors must provide clear statistical analysis for the observed changes.

Reply: Following the suggestion, we have provided the average m⁶A signal density by boxes. As shown in New Appendix Figure S3B, EPZ5676 treatment can significantly downregulate the m⁶A modification at TWINS regions.

Figure legend: Bar chart comparing the TWINS-region coverage of MLL-r targets in MeRIP-seq and GLORI with and without EPZ5676 treatment. The *P* value was analyzed by Proportion Test.

4. The IPs in the Supp Fig 5b has a very high background in the IgG control and make it difficult to interpret. Please provide a better IP.

Reply: Thank you for pointing this out. Following the suggesting, we have redone the IP. The new figure has been added in the revised manuscript.

5. Indicate the additional band in the Supp Fig5f. Unspecific? Light chain?

Reply: Thank you for pointing this out. The additional band in the Supp Fig5f is a unspecific band, and has labelled "*" on the figure and stated in the figure legend.

New Appendix Figure S5F

6.To enhance reader comprehension and provide better context for the study's findings, it would be beneficial to include a small paragraph in the Introduction outlining what is currently known about RBM15. This would help to clearly establish the foundation upon which the current research builds and highlights the significance of identifying RBM15B's novel role.

Reply: We appreciate your comment. Following the suggestion, we have outlined the current researches on RBM15B in the introduction section.

7.Please provide an alignment of Menin finger domain and K79HMBD. It is unclear how you identified the residues.

Reply: As suggested, we have provided the alignment of Menin finger domain and K79HMBD in the material method. In brief, we first identified the Domain Boundary. The finger domain of Menin (UniProt: O00255) comprises residues 400–610 and adopts an antiparallel β -sheet fold that forms a histone-binding cradle. Structural analysis of the Menin-H3K79me2 complex (PDB: 4GQ4) reveals that this domain directly engages the H3K79me2 peptide through a conserved basic pocket formed by residues R476, K480, and K483, which coordinate the methylated lysine via hydrogen bonding and hydrophobic interactions (Lin et al, 2023). For RBM15B, residues 1-138 were designated as the K79me2-binding domain (K79HMBD). Sequence alignment of human Menin and RBM15B was performed using Clustal Omega (<https://www.ebi.ac.uk/Tools/msa/clustalo/>), focusing on the defined domain boundaries. The designation K79HMBD for RBM15B residues 1-138 reflects: (i) Exclusive binding specificity to H3K79me2; (ii) Functional distinction from N-terminal RNA recognition motifs (RRMs) responsible for RNA binding.

The structural validation and docking methodology as follows: Protein structures were retrieved from the AlphaFold database(Abramson et al, 2024). Domain

boundaries were mapped using AlphaFold-predicted structures and cross-referenced with Pfam functional domains. Prior to docking, structures were processed in PyMOL 2.5.3 to remove low-confidence regions. Global rigid docking was performed using ZDOCK 3.0.2 to predict H3K79me2 binding modes. Resultant models were visualized and analyzed in PyMOL 2.5.3. We have added this description into the revised manuscript (**page 31, lines 881-898**).

8.The labels are missing for the panel 4a.

Reply: We have labelled this information “si-NC, si-RBM15B-1 or -2” in the panel 4a. Thank you.

9.For the polysome profiling, it is common to show two conditions (Control vs KD) on the same graph to compare overall protein synthesis. Please merge the two panels in Fig 5f as well as in Fig 5g.

Reply: We agree and have merged the two panels (**New Appendix Fig.S9A-C**).

References:

- Abramson J, Adler J, Dunger J, Evans R, Green T, Pritzel A, Ronneberger O, Willmore L, Ballard AJ, Bambrick J et al (2024) Accurate structure prediction of biomolecular interactions with AlphaFold 3. *Nature* 630: 493-500
- Cheng Y, Xie W, Pickering BF, Chu KL, Savino AM, Yang X, Luo H, Nguyen DT, Mo S, Barin E et al (2021) N(6)-Methyladenosine on mRNA facilitates a phase-separated nuclear body that suppresses myeloid leukemic differentiation. *Cancer Cell* 39: 958-972 e958
- Corovic M, Hoch-Kraft P, Zhou Y, Hallstein S, Konig J, Zarnack K (2025) m(6)A in the coding sequence: linking deposition, translation, and decay. *Trends Genet* S0168-9525: 132-135
- Erb MA, Scott TG, Li BE, Xie H, Paulk J, Seo HS, Souza A, Roberts JM, Dastjerdi S, Buckley DL et al (2017) Transcription control by the ENL YEATS domain in acute leukaemia. *Nature* 543: 270-274
- Erb MA, Scott TG, Li BE, Xie H, Paulk J, Seo HS, Souza A, Roberts JM, Dastjerdi S, Buckley DL et al (2017) Gene Expression Omnibus GSE GSE82116 (<https://www.ncbi.nlm.nih.gov/geo/query/acc.cgi?acc=GSE82116>). [DATASET]
- Gilan O, Lam EY, Becher I, Lugo D, Cannizzaro E, Joberty G, Ward A, Wiese M, Fong CY, Ftouni S et al (2016) Functional interdependence of BRD4 and DOT1L in MLL leukemia. *Nat Struct Mol Biol* 23: 673-681
- Gilan O, Lam EY, Becher I, Lugo D, Cannizzaro E, Joberty G, Ward A, Wiese M, Fong CY, Ftouni S et al (2016) Gene Expression Omnibus GSE GSE71780 (<https://www.ncbi.nlm.nih.gov/geo/query/acc.cgi?acc=GSE71780>). [DATASET]

- Janssens DH, Meers MP, Wu SJ, Babaeva E, Meshinchi S, Sarthy JF, Ahmad K, Henikoff S (2021) Automated CUT&Tag profiling of chromatin heterogeneity in mixed-lineage leukemia. *Nat Genet* 53: 1586-1596
- Janssens DH, Meers MP, Wu SJ, Babaeva E, Meshinchi S, Sarthy JF, Ahmad K, Henikoff S (2021) Gene Expression Omnibus GSE159608 (<https://www.ncbi.nlm.nih.gov/geo/query/acc.cgi?acc=GSE159608>). [DATASET]
- Kramer MH, Zhang Q, Sprung R, Day RB, Erdmann-Gilmore P, Li Y, Xu Z, Helton NM, George DR, Mi Y et al (2022) Proteomic and phosphoproteomic landscapes of acute myeloid leukemia. *Blood* 140: 1533-1548
- Li S, Liu Y, Wu X, Pan M, Zhao H, Hong Y, Zhang Q, Hu S, Ouyang A, Li G et al (2025) The m(5)C methyltransferase NSUN2 promotes progression of acute myeloid leukemia by regulating serine metabolism. *Cell Rep* 44: 115661
- Lin J, Wu Y, Tian G, Yu D, Yang E, Lam WH, Liu Z, Jing Y, Dang S, Bao X et al (2023) Menin "reads" H3K79me2 mark in a nucleosomal context. *Science* 379: 717-723
- Lourenco C, Resetca D, Redel C, Lin P, MacDonald AS, Ciaccio R, Kenney TMG, Wei Y, Andrews DW, Sunnerhagen M et al (2021) MYC protein interactors in gene transcription and cancer. *Nat Rev Cancer* 21: 579-591
- Meyer KD (2019) m(6)A-mediated translation regulation. *Biochim Biophys Acta Gene Regul Mech* 1862: 301-309
- Patil DP, Chen C-K, Pickering BF, Chow A, Jackson C, Guttman M, Jaffrey SR (2016) m(6)A RNA methylation promotes XIST-mediated transcriptional repression. *Nature* 537: 369-373
- Rogalska ME, Mancini E, Bonnal S, Gohr A, Duniak BM, Arecco N, Smith PG, Vaillancourt FH, Valcarcel J (2024) Transcriptome-wide splicing network reveals specialized regulatory functions of the core spliceosome. *Science* 386: 551-560
- Stein EM, Garcia-Manero G, Rizzieri DA, Tibes R, Berdeja JG, Savona MR, Jongen-Lavrenic M, Altman JK, Thomson B, Blakemore SJ et al (2018) The DOT1L inhibitor pinometostat reduces H3K79 methylation and has modest clinical activity in adult acute leukemia. *Blood* 131: 2661-2669
- Yi Y, Ge S (2022) Targeting the histone H3 lysine 79 methyltransferase DOT1L in MLL-rearranged leukemias. *J Hematol Oncol* 15: 35

Dear Dr Chen,

Thank you again for transferring your amended manuscript (EMBOJ-2025-121070) to The EMBO Journal, together with a point-by-point response to the issues raised during peer-review at the previous venue. Please accept my sincere apologies for the unusual protraction with the reassessment due to delayed expert input and detailed discussion in the editorial team. Your revised study was sent back to the referees for their scientific reassessment, and we have received detailed re-reports from one of them, which I enclose below. Please note that while referee #1 was at this time not able to look back into your study, we have editorially considered your response to his/her concerns and found them to be addressed satisfactorily. As you will see, referee #2 states that the work has been substantially enhanced by the revisions and s/he is now in favour of publication, pending minor revision.

Thus, we are pleased to inform you that your manuscript has been accepted in principle for publication in The EMBO Journal.

Please carefully consider the remaining minor points raised by referee #2 by adding complementary data and quantification, or introducing caveats into the manuscript text where appropriate.

Also, we now need you to take care of a number of minor issues related to formatting and data annotation, which I will share shortly in a separate message, together with additional changes and requests by our production team for Source Data provision.

Please submit a revised version of the manuscript using the link enclosed below, addressing the advisor's comments.

Thank you again for giving us the chance to consider your manuscript for The EMBO Journal, I look forward to hearing from you and receiving your final revised version of the manuscript.

Kind regards,

Daniel Klimmeck

>> Please add up to five keywords to your study.

>> Author Contributions: Remove the author contributions information from the manuscript text. Note that CRediT has replaced the traditional author contributions section as of now because it offers a systematic machine-readable author contributions format that allows for more effective research assessment. and use the free text boxes beneath each contributing author's name to add specific details on the author's contribution.

More information is available in our guide to authors.
<https://www.embopress.org/page/journal/14602075/authorguide>

>> Section order should be as follows: title page with complete author information, abstract, keywords, introduction, results, discussion, methods, data availability section, acknowledgements, disclosure and competing interests statement, references, main figure legends, tables, expanded figure legends.

>> Figure callouts: There is a citation left for a "Supplementary Table 4", please correct.

>> Funding: please enter the following funding information in the list of funders in our online system: " National Natural Science (32400439), Guangdong Province (No.2021B1515020002, 2019JC05N394, 2022A1515140018), and Guangzhou (2024A04J5004), and Fundamental Research Funds for the Central Universities, Sun Yat-sen University (No. 231gbj010)".

>>Appendix file with ToC: please upload the final version as a PDF with the blue font removed.

>> Data availability section: remove the referee token and make sure the GEO dataset is made publicly accessible.

>> Author Checklist: Data availability>>primary datasets section: remove the referee token.

>> Consider additional changes and comments from our production team as indicated below:

- Figure legends:

1. Please note that the exact p values are not provided in the legends of figures 1A, B, D, E, L; 2F, G, N; 3E, 4A, C, D, E, F, G, I J; 5B, C, F, G, L M; 6A, C, D, F, I, J; 7A, E, D.

2. Please note that the box plots need to be defined in terms of minima, maxima, centre, bounds of box and whiskers, and percentile in the legend of figure 1A, B

3. Please note that the error bars are not defined in the legend of figure 1D.

Referee #2:

The authors have made a commendable effort in performing additional experiments to address my prior concerns. While these new data improve the manuscript, there are still a few important issues that need to be reconsidered before the study can be accepted for publication.

My main concern is about the GFP-based reporter assay and the assessment of translational efficiency (TE).

1. The current approach quantifies GFP-positive cells as a measure of TE. However, mean fluorescence intensity (MFI) would provide a more accurate assessment of translational output. To enable a proper comparison between WT and mutant TWIN sequences, the authors should also quantify RNA levels from each reporter and normalize GFP-MFI to these values. In addition, as the WT and mutant reporters are stably expressed in distinct cell lines, comparisons across these lines may not fully reflect direct effects of the sequence differences.

2. MFI quantification is also particularly important in the context of RBM15B overexpression and knockdown experiments, where both WT and mutant HOXA9-TWIN reporters appear significantly affected. It would be informative to determine whether these effects remain significant when analyzed by MFI. If so, the authors should clarify how RBM15B overexpression could influence the mutant reporter, which would not be expected to respond in the same manner as the WT construct.

3. Regarding the two 5'UTRs tested, HOXA9 and MYC, it appears that the HOXA9 reporter was successfully transduced, whereas the MYC construct was expressed in only a very small subset of cells (2-3%). To strengthen these experiments, the authors may consider either repeating the transduction to increase the proportion of GFP-positive cells and then quantifying MFI, or alternatively, testing transient expression of the constructs if technically feasible.

In summary, addressing these points will substantially improve the rigor and interpretability of the findings, thereby strengthening the overall conclusions of the manuscript.

Point-by-point response to editor's and reviewers' comments

Editor's comments:

Thank you again for transferring your amended manuscript (EMBOJ-2025-121070R1) to The EMBO Journal, together with a point-by-point response to the issues raised during peer-review at the previous venue.

We are pleased to inform you that your manuscript has been accepted in principle for publication in The EMBO Journal. Please carefully consider the remaining minor points raised by referee #2 by adding complementary data and quantification, or introducing caveats into the manuscript text where appropriate. Also, we now need you to take care of a number of minor issues related to formatting and data annotation, which I will share shortly in a separate message, together with additional changes and requests by our production team for Source Data provision.

Please submit a revised version of the manuscript using the link enclosed below, addressing the advisor's comments.

Reply: We appreciate all the efforts and your decision, thank you very much. In response to your suggestions, we have carefully addressed all minor issues from referee #2 about adding the complementary data and quantification, and introducing caveats into the manuscript text where appropriate; We have also addressed the editorial issues related to formatting and data annotation, and Source Data. Specific revision included:

- (1) We have normalized GFP-MFI to the relative RNA levels (GFP-MFI / relative expression of mRNAs, GFP-MFI / RE-RNAs) and reassess the translational efficiency (TE) of GFP.
- (2) As the suggestion, we have repeated the transduction of MYC construct and calculated the GFP-MFI/ RE-RNAs to compare the translational output again.
- (3) 4 keywords have been added in the study.
- (4) Following the request, we have removed the referee token and release the data availability publicly accessible. The change has also updated in 'Author Checklist' file.
- (5) Following the request, we have added the exact *P* values in the legends of figures 1A, B, D, E, L; 2F, G, N; 3E, 4A, C, D, E, F, G, I J; 5B, C, F, G, L M; 6A, C, D, F, I, J; 7A, E, D.
- (6) we have defined the box plots in the legend of figure 1A, B. The Error bars denote means \pm SEM in figure 1D.
- (7) We have uploaded the final version of as Appendix file a PDF without the blue font. The "Supplementary Table 4" is corrected to "Appendix Table S4".
- (8) We changed and added the funding information in the online system. We have removed the "Author Contributions" section and followed the "CRedit" as suggested.

The input from you and the colleagues has helped greatly improve our manuscript. Thank you.

Referee #2 Comment:

The authors have made a commendable effort in performing additional experiments to address my prior concerns. While these new data improve the manuscript, there are still a few important issues that need to be reconsidered before the study can be accepted for publication.

My main concern is about the GFP-based reporter assay and the assessment of translational efficiency (TE).

1. The current approach quantifies GFP-positive cells as a measure of TE. However, mean fluorescence intensity (MFI) would provide a more accurate assessment of translational output. To enable a proper comparison between WT and mutant TWIN sequences, the authors should also quantify RNA levels from each reporter and normalize GFP-MFI to these values. In addition, as the WT and mutant reporters are stably expressed in distinct cell lines, comparisons across these lines may not fully reflect direct effects of the sequence differences.

Reply: We appreciate your insightful comment. As suggested, we have provided the mean fluorescence intensity (MFI) to reassess the GFP translational output. In addition, we have also performed the qRT-PCR to detect the relative GFP mRNA levels in the cell lines with WT or mutant reporters. The results showed slight changes of GFP mRNA levels between the cells with WT and mutant reporters (**New Appendix Figure S10B**). Following the suggestion, we have normalized GFP-MFI to the relative RNA levels (GFP-MFI / relative expression of mRNAs, GFP-MFI / RE-RNAs) and reassess the translational output. The results have also showed that upregulation of RBM15B significantly enhanced the GFP-MFI / RE-RNAs in *TWINS^{m6Awt}-GFP* cells, while downregulation of RBM15B reduced it. However, modulating RBM15B expression had a limited influence on GFP-MFI / RE-RNAs in *TWINS^{m6Amut}-GFP* cells (**Appendix Figure S10**). Indeed, the method you suggested conducts a more accurate quantitative analysis of the differences of GFP translation efficiency through normalization of GFP-MFI to relative RNA levels, effectively avoiding interference from the RNA transcription differences between stably transfected cells. We are grateful for this suggestion. The new data and analysis have been added in the revised manuscript (**page 15, lines 419-426; page 25, lines 724-731**).

2. MFI quantification is also particularly important in the context of RBM15B overexpression and knockdown experiments, where both WT and mutant HOXA9-TWIN reporters appear significantly affected. It would be informative to determine whether these effects remain significant when analyzed by MFI. If so, the authors should clarify how RBM15B overexpression could influence the mutant reporter, which

would not be expected to respond in the same manner as the WT construct.

Reply: We appreciate your comment. Following your suggesting, we have normalized the GFP-MFI to the relative RNA levels. The GFP-MFI / RE-RNAs showed that RBM15B overexpression and knockdown regulate the translational output of WT HOXA9-TWIN reporters (**New Figure 5L and M**). And RBM15B had no significant influence on the translational output of mutant HOXA9-TWIN reporters. GFP-MFI / RE-RNAs can quantify the differences of GFP translation efficiency more accurate among stably transfected cells. Very good commend indeed. Thank you again.

3. Regarding the two 5'UTRs tested, HOXA9 and MYC, it appears that the HOXA9 reporter was successfully transduced, whereas the MYC construct was expressed in only a very small subset of cells (2-3%). To strengthen these experiments, the authors may consider either repeating the transduction to increase the proportion of GFP-positive cells and then quantifying MFI, or alternatively, testing transient expression of the constructs if technically feasible.

Reply: We thank you for the suggestion. As suggested, we have repeated the transduction of MYC construct and calculated the GFP-MFI/ RE-RNAs to compare the translational output again. In the assay, we found a significant improvement of MFI in the MYC-TWIN reporters (**right in Appendix Figure S10A**).

Similar to their effects on HOXA9-TWIN reporters, both RBM15B manipulation and EPZ5676 treatment significantly modulated GFP-MFI/RE-RNAs in WT MYC-TWIN reporters. No significant effect was observed in the mutant reporters. (**New Appendix Figure S10E, I, J, and S11D, and E**).

Thank you for these comments to strengthen the overall conclusions of the manuscript.

Dear Dr Chen,

Thank you for submitting the revised version of your manuscript. I have now evaluated your amended manuscript and concluded that the remaining minor concerns have been sufficiently addressed.

I am thus pleased to inform you that your manuscript has been accepted for publication in the EMBO Journal.

Best regards,

Daniel Klimmeck

Daniel Klimmeck, PhD
Senior Editor
The EMBO Journal
EMBO
Postfach 1022-40
Meyerhofstrasse 1
D-69117 Heidelberg
contact@embojournal.org

Please note that it is The EMBO Journal policy for the transcript of the editorial process (containing referee reports and your response letters) to be published as an online supplement to each paper. If you should prefer removal of any referee-only figures included in the point-by-point response(s), e.g. because they may still be used for future publication or because they have been reproduced from published work by others, please do let us know immediately via response email.